# Leading countries in global science increasingly receive more citations than other countries doing similar research

Charles J. Gomez[1]✉, Andrew C. Herman[2] and Paolo Parigi[3]

Citations and text analysis are both used to study the distribution and flow of ideas between researchers, fields and countries, but the resulting flows are rarely equal. We argue that the differences in these two flows capture a growing global inequality in the production of scientific knowledge. We offer a framework called 'citational lensing' to identify where citations should appear between countries but are absent given that what is embedded in their published abstract texts is highly similar. This framework also identifies where citations are overabundant given lower similarity. Our data come from nearly 20 million papers across nearly 35 years and 150 fields from the Microsoft Academic Graph. We find that scientific communities increasingly centre research from highly active countries while overlooking work from peripheral countries. This inequality is likely to pose substantial challenges to the growth of novel ideas.

International inequalities have the potential to disrupt the generation and flow of knowledge in the global scientific community. Research can make an impact only if it is visible. Of course, journals and scientific reputations shape the visibility of research in conjunction with the overall quality of a study. But so too do national scientific infrastructures and reputations. The scale and quality of research, and even library subscription practices, are heavily influenced by the amount of funding countries set aside for the scientific enterprise[1–4]. Countries with these advantages probably receive additional citations for their research, over and above what one would expect just from the subject matter of that research[5], something that scholars of the Global South have suggested before[2,6,7]. Studying these inequalities systematically is difficult because the many relevant factors are so deeply intertwined. In this paper, we introduce a framework we call citational lensing that takes advantage of the strong relationship between citations and textual similarity to identify countries that receive more (or fewer) citations than one would expect if citations and textual similarity were perfectly aligned.

Much like how gravity distorts our perception of light, national factors distort our perception of international science. The intuition behind our approach is to think of international science as a multiplex network, with the citations running between countries representing one type of connection, and the textual similarity of their research output representing another. Citational lensing is measured as the difference between the weighted edges that correspond to international citations and the weighted edges that correspond to textual similarity. The result is a set of connections that represent how much more one country cites another, relative to what we would expect if citations and textual similarity reflected each other perfectly. We call this layer of the multiplex network the citational well. Filtering out the effect of other factors is also possible, leaving distortion captured in the citational well to represent a more narrowly defined notion of inequality. This is shown in Fig. 1, where we represent science as a multiplex network with three layers: $\mathbf{L}_{\text{citation}}$ is the citation network between countries containing the citation flow from country $i$ to country $j$, $\mathbf{L}_{\text{text}}^{\text{T}}$ is the relative similarity of the text of country $j$'s research output to that of country $i$ by applying a Kullback–Leibler divergence (KLD) measure[8] to the distinct national signatures of countries $i$ and $j$ produced by a supervised topic model called a labelled latent Dirichlet allocation (LDA) model[9], and $\mathbf{L}_{\text{distortion}}$ is what we call the citational well defined as the difference between the two layers:

$$\mathbf{L}_{\text{distortion}} = \mathbf{L}_{\text{citation}} - \mathbf{L}_{\text{text}}^{\text{T}} \tag{1}$$

$\mathbf{L}_{\text{distortion}}$ represents the distortion in the citations from country $i$ to country $j$, relative to what we would expect given the similarity of the research written by scientists in these two countries.

Our approach builds on the long-standing tradition in the science of science that uses citation networks and text analysis of scientific papers to embody the flow of ideas in science and map its structure, as well as the distribution and spread of knowledge within it[5,10–14]. Yet the citation networks and the textual similarity between fields are not always aligned. There are commonly more citations between fields than we would expect on the basis of the textual similarity of their papers, or conversely, more similarity in the text than we would expect given the number of citations flowing between those fields[12,15].

It is not clear whether this misalignment between citations and textual similarity has any substantive importance for scientific fields[16]. Bibliometrics has treated the issue as a question of ground truth, where the differences between the two are less important than their respective differences from a third, external criterion[17,18]. In this line of thinking, the misalignment of citations and textual similarity is simply beside the point and does not impact the larger goal of mapping science. In the science of science, meanwhile, the misalignment is taken as a sign that any model of diffusion or communication between scientific fields needs to take both citations and textual similarity into account[11,12,15].

When it comes to countries, misalignments between citations and textual similarity carry practical significance. This is because

[1]Department of Sociology, Queens College, City University of New York, New York, NY, USA. [2]Department of Sociology, University of California, Los Angeles, CA, USA. [3]Institute for Social Science Research (IRiSS) at Stanford University, Stanford, CA, USA. ✉e-mail: charles.gomez@qc.cuny.edu

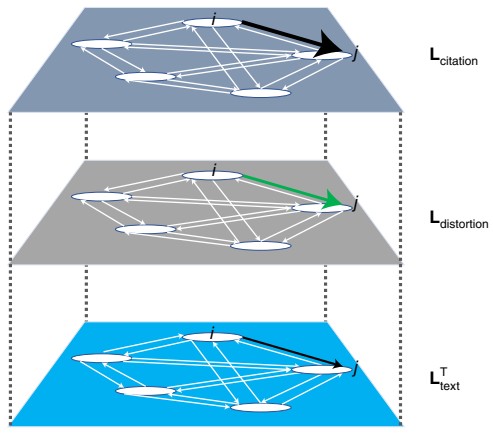

**Fig. 1 | The construction of the multiplex network, the citational well.**
The citational well is the result of subtracting the edge weights of the
international text similarity network from the corresponding edge weights
of the international citation network.

they represent the combined effect of several factors, including
the overall quality of research and national reputations. The cita-
tional well provides a noisy signal that can be further refined to
better approximate specific factors, as we show with respect to
national reputation.

Identifying citational distortions is critical to discussions on
inequality in global scientific knowledge production. Knowledge
production is overwhelmingly skewed towards resource-wealthy
countries such as the United States and those in Western Europe
and East Asia (to name a few) that house the best universities,
Nobel winners and journal editors, so identifying undercited coun-
tries both promotes the inclusion of often-excluded voices and
helps foster the scientific enterprises of these countries. The type of
distortion we consider here is also likely to be problematic for sci-
entific progress if knowledge remains unincorporated and human
capital unused.

## Results

To illustrate how citational lensing can be applied, we use roughly
20 million academic papers in nearly 150 fields and subfields
from 1980 to 2012 in the Microsoft Academic Graph (MAG), one
of the most extensive metadata repositories of academic publica-
tions. These data include metadata such as citations, along with the
abstract text of published research articles. We show how citational
lensing can be used to characterize changes in the international
scientific hierarchy over time and how it can be scaled to cover all
of science.

**Citations and recognition.** In Fig. 2a, we count for each year the
number of countries that are present in both the international text
similarity network and the international citation network for each
field and for each year from 1980 to 2012. The plot captures the
distribution of fields in each year, as well as the overall average
number of countries represented in fields for each year. Across all
types of fields, the number of countries represented in the global
scientific conversation is increasing, as captured by journal arti-
cles. Note that while Fig. 2a shows steady growth in the number
of countries in the international scientific community over time,
this appears to taper off in the years immediately prior to 2012. In
Supplementary Tables 3 and 4, we test how much variance in our
citational distortion and thus our text similarity measures is due to
the result of the sheer volume of papers produced by authors from
countries using hierarchical linear models. The mediating impact

of $N$ papers is modest. Adjusting for $N$ papers reduces the vari-
ance in country-related text similarity by 19%, from 0.16 to 0.13.
Likewise, adjusting for $N$ papers reduces the country-related vari-
ance in citational distortion by 12.5%, from 0.08 to 0.07.

While there may be more countries participating in global sci-
ence, the extent to which their work is visible and valued may still be
highly stratified. The relationship between the textual similarity of
countries' research output and the number of citations they receive
is one simple way to think about this. In a world where all research
is equally visible and equally valued, we would expect an extremely
close relationship between the similarity of research in their text and
the citations that flow between them. We test the extent to which
$\mathbf{L}_{citation}$ corresponds with $\mathbf{L}_{text}$ across fields and over time in Fig. 2b.
This is done using a form of network regression model called the
semi-partialing quadratic assignment procedure (QAP)[19] to capture
the extent to which citations are associated with $\mathbf{L}_{text}^{T}$. Specifically,
we run two types of QAP models based on which country nodes are
included in $\mathbf{L}_{text}$ and $\mathbf{L}_{citation}$, the first using all countries present in the
data that year and the second using only 'core' scientific countries—
specifically, those in Western Europe and East Asia (that is, China,
Japan and South Korea), alongside the United States, Canada,
Australia, New Zealand, Singapore and Israel. (In Supplementary
Table 2, we itemize every country we use in our analysis, parsed by
their core and periphery classification.) These countries are com-
monly considered to be the leaders in scientific research (housing
the best universities and most-cited scientists, publishing work in
leading outlets and so on) and as such form the core of global sci-
entific communities where most scientific activity takes place[20,21].

We run each type of QAP model for every field in every year,
from 1980 to 2012. The dependent variable in each model is the
network represented in $\mathbf{L}_{text}^{T}$ for that field and year, while the inde-
pendent variable is the international citation network for that field
and year, given as $\mathbf{L}_{citation}$. We thus regress how similar country $j$ is to
country $i$ in $\mathbf{L}_{text}^{T}$ on how much country $j$ cites country $i$. There are
two 'grand average' trends in Fig. 2b: the average $\beta$ coefficient value
across all fields in each year, and the shaded area in the plot, which
is the grand standard errors across all $\beta$s in each year. Note that
we only consider $\beta$s that are statistically significant at a two-tailed
$P$ value threshold of 0.05.

There are a few notable take-aways from these trends. A
one-standard-deviation increase in the number of citations, as
defined for each QAP model in the year 2012, is associated with
0.228-standard-deviation-higher KLD scores in $\mathbf{L}_{text}$ among all
countries (that is, an all-inclusive model with both core and periph-
ery countries) ($N = 135$; 95% confidence interval (CI), (0.226,
0.231); two-tailed $t$-tests). However, when we compare the QAP
model with just core countries against the QAP model that includes
all countries (that is, core and periphery), we find that citations
have a consistent and strong relationship with the similarity of lan-
guage in international research, where a one-standard-deviation
increase in the number of citations in 2012 is associated with
0.312-standard-deviation-higher KLD scores in $\mathbf{L}_{text}$ for just core
countries ($N = 139$; 95% CI, (0.309, 0.315); two-tailed $t$-tests). This
is also evidenced by the overall higher average $\beta$ values that remain
fairly steady over time, albeit with a slight increase in the same
period within the core-country QAP model. The average $\beta$ value for
the core–periphery model, by contrast (which reflects the relation-
ship across all countries in the dataset), is not only lower overall than
that for the core country model but is also weakening over time.
(In Supplementary Figs. 8 and 9, we rerun the QAP models to now
include countries' $\beta$ coefficients that include only peripheral coun-
tries, similar to the core network that contains only core countries.)

Figure 3 plots the average citational distortion (that is, the aver-
age in-degree centrality) that each country experiences in $\mathbf{L}_{distortion}$.
This should be interpreted as the difference in the number of stan-
dard deviations between edges in the citation network (measured in

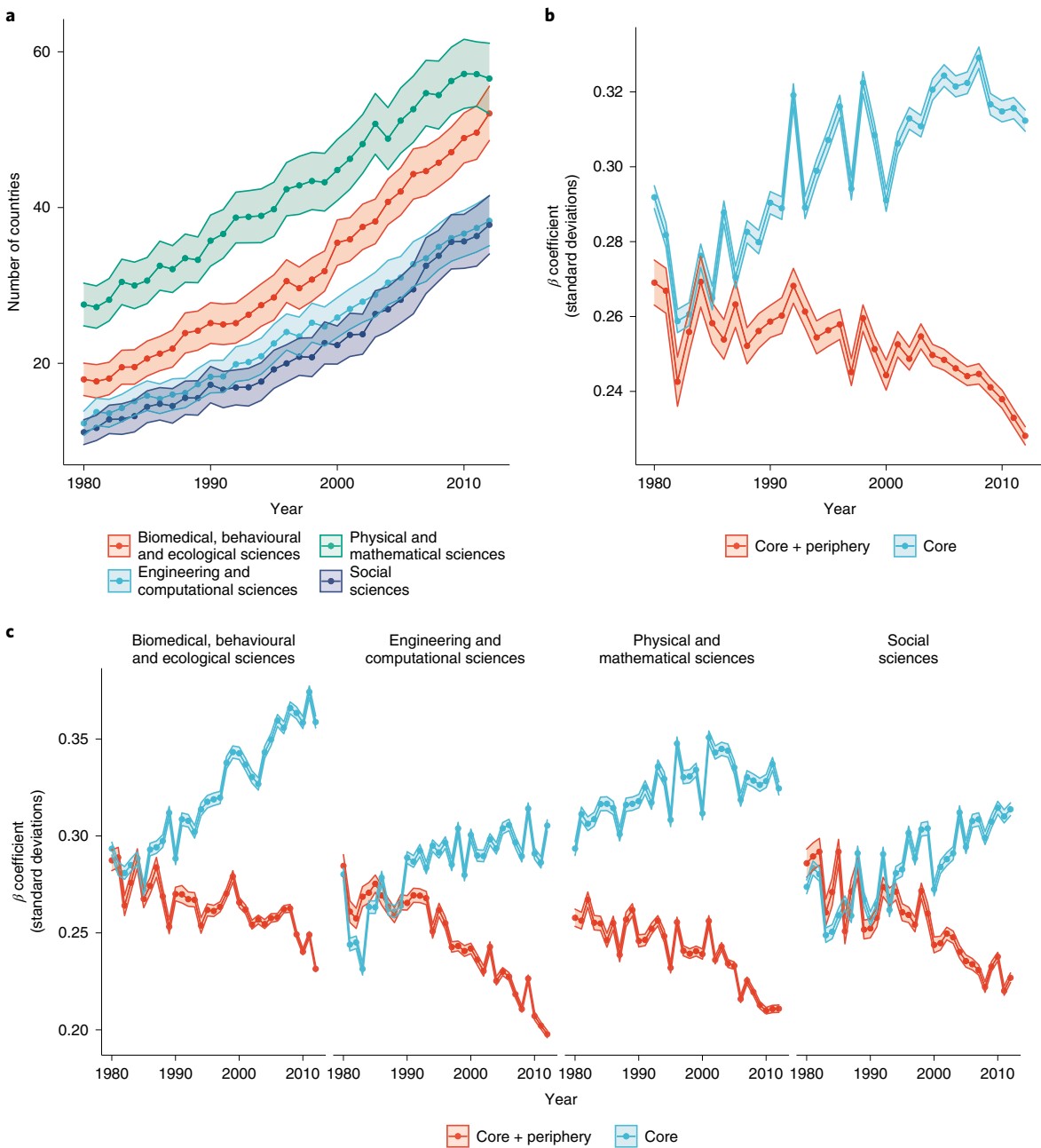

**Fig. 2 | Relationship between citations and text over time. a**, The average number of countries present in both the international text similarity network and the international citation network over time from 1980 to 2012, with trend lines for each field type. **b**, The average of statistically significant $\beta$ coefficients from each field's yearly QAP model for citations plotted on the $y$ axis and over time on the $x$ axis from 1980 to 2012. The shading around the trends denotes the standard errors across fields. **c**, The trends plotted in **b**, but parsed by the type of field. The shading around the trends denotes the grand standard errors across $\beta$ coefficients.

terms of standard deviations relative to the citation network) and those in the text similarity network (measured in terms of standard deviations relative to the text similarity network). Figure 3a highlights many of the countries with the greatest positive distortions. Figure 3b shows the average citational distortion over time among core countries and among periphery countries. (As the United States is an outlier, we remove it from the average core trend in Fig. 3b and all subsequent in-degree plots.) The United States is the most central country in the citation networks, for all fields and over time. The deviance of the United States' centrality in $L_{distortion}$ suggests that it is, on average, highly overcited. This holds true for other power players in global science, such as Germany, the Netherlands,

the United Kingdom and Japan. The other major trend is that China rises considerably over the past few decades, from being undercited throughout the 1980s and early 1990s to being overcited in the 2000s, even quickly approaching many countries in Western Europe. As shown in Fig. 3b, the gap between core and periphery countries is growing substantially over time, as core countries are increasingly overcited for their work relative to what they study, while peripheral countries are increasingly undercited for their work. (As the growing gap between the trends in core and periphery countries may be the result of the number of countries, in Supplementary Figs. 28 and 29, we censor them on the basis of when countries first appeared in the data for each field, finding that our results still hold.)

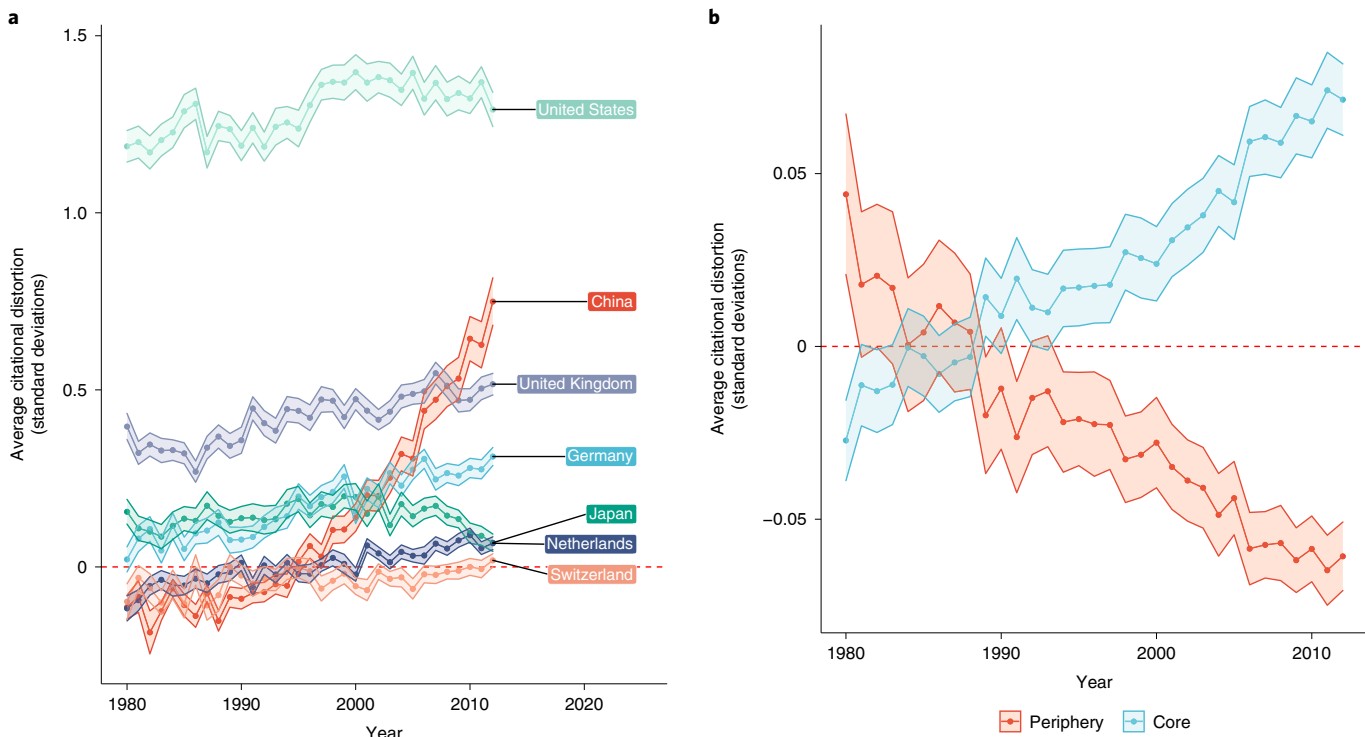

**Fig. 3 | Comparing global citational distortion over time. a**, The average national citational distortion in **L**$_{distortion}$ plotted across fields on the y axis and over time on the x axis for select countries. **b**, The average national citational distortion plotted for core countries and peripheral countries on the y axis and over time on the x axis. The gap in the average distortion in the citational well is growing between core and periphery countries. The shading around the trends denotes the standard errors of these averages. Note that citational distortion should be interpreted as the difference in the number of standard deviations between edges in the citation network (measured in terms of standard deviations relative to the edge weights in the citation network) and those in the text similarity network (measured in terms of standard deviations relative to the edge weights in the text similarity network).

Figure 4a expands the plots in Fig. 3a, while Fig. 4b expands the plots in Fig. 3b, but the trends are now averaged by research area. While the most overcited countries are still present across fields, the ordering differs. The United States remains the most central across all fields. China, however, has overtaken some European countries in the physical and mathematical sciences and in the engineering and computational sciences. In the biomedical, behavioural and ecological sciences, China is nearing Germany but trailing the United Kingdom. The exception is the social sciences, where China is on par with Germany but noticeably trails the United Kingdom. When comparing core and periphery countries in Fig. 4b by research area, the gap is most pronounced in the physical and mathematical sciences, followed by the engineering and computational sciences and the biomedical, behavioural and ecological sciences. The gap has only recently emerged in the social sciences.

**A stagnating hierarchy.** Thus far, we have focused on the most overcited distorted countries across different fields and over time. However, most countries are undercited. With some exceptions, countries that are overcited or undercited remain constant over time. To unpack this, Fig. 5 plots the average distortion for several countries at two points in time—the year 2000 on the x axis and the year 2012 on the y axis—but parsed by research area and in four transnational regions: (1) Europe, (2) Asia, (3) Africa and the Middle East and (4) Latin America and the Caribbean. First, without any divides by research area or transnational region as shown here, all countries cluster very closely near the y = x parity line (with a Pearson's product–moment correlation of $\rho = 0.659$; $t = 9.59$; d.f. = 120; 95% CI, (0.545, 0.749); $P < 2.2 \times 10^{-16}$). Countries tend to cluster in one of two quadrants: overcited in both 2000 and 2012

(that is, data points that are positive in both years) or undercited in both 2000 and 2012 (that is, data points that are negative in both years). Overcited countries are typically part of the core of global science. There are far fewer countries in the overcited quadrant than in the undercited quadrant, which comprises the periphery of global science. Except the power players in global science, most countries seem to be under-recognized for their work. In other words, most countries remain in either the lower left quadrant or the upper right quadrant, indicating that countries do not generally change their station in their distortion over time.

That being said, note as well that some regions and research areas are more stable than others in terms of the average citational distortion for countries across the two periods. Unsurprisingly, Europe shows high correlations for all research areas ($\rho = 0.68$; $t = 5.01$; d.f. = 29; 95% CI, (0.431, 0.834); $P < 2.47 \times 10^{-5}$ for biomedical, behavioural and ecological sciences; $\rho = 0.66$, $t = 5.00$; d.f. = 32; 95% CI, (0.418, 0.817); $P < 1.99 \times 10^{-5}$ for engineering and computational sciences; $\rho = 0.76$; $t = 7.13$; d.f. = 37; 95% CI, (0.586, 0.868); $P < 1.92 \times 10^{-8}$ for physical and mathematical sciences; $\rho = 0.66$; $t = 4.27$; d.f. = 23; 95% CI, (0.366, 0.839); $P < 2.89 \times 10^{-4}$ for social sciences). Other regions show more variation across research areas. While citational distortions in engineering and computational sciences are highly correlated in Asia, with lower correlations in other research areas, Latin America and the Caribbean show a different pattern, with a high correlation in the biomedical, behavioural and ecological sciences and lower correlations in other research areas. By contrast, Africa and the Middle East have a lower (albeit statistically significant) correlation in the engineering and computational sciences, but lower and non-significant correlations in other research areas.

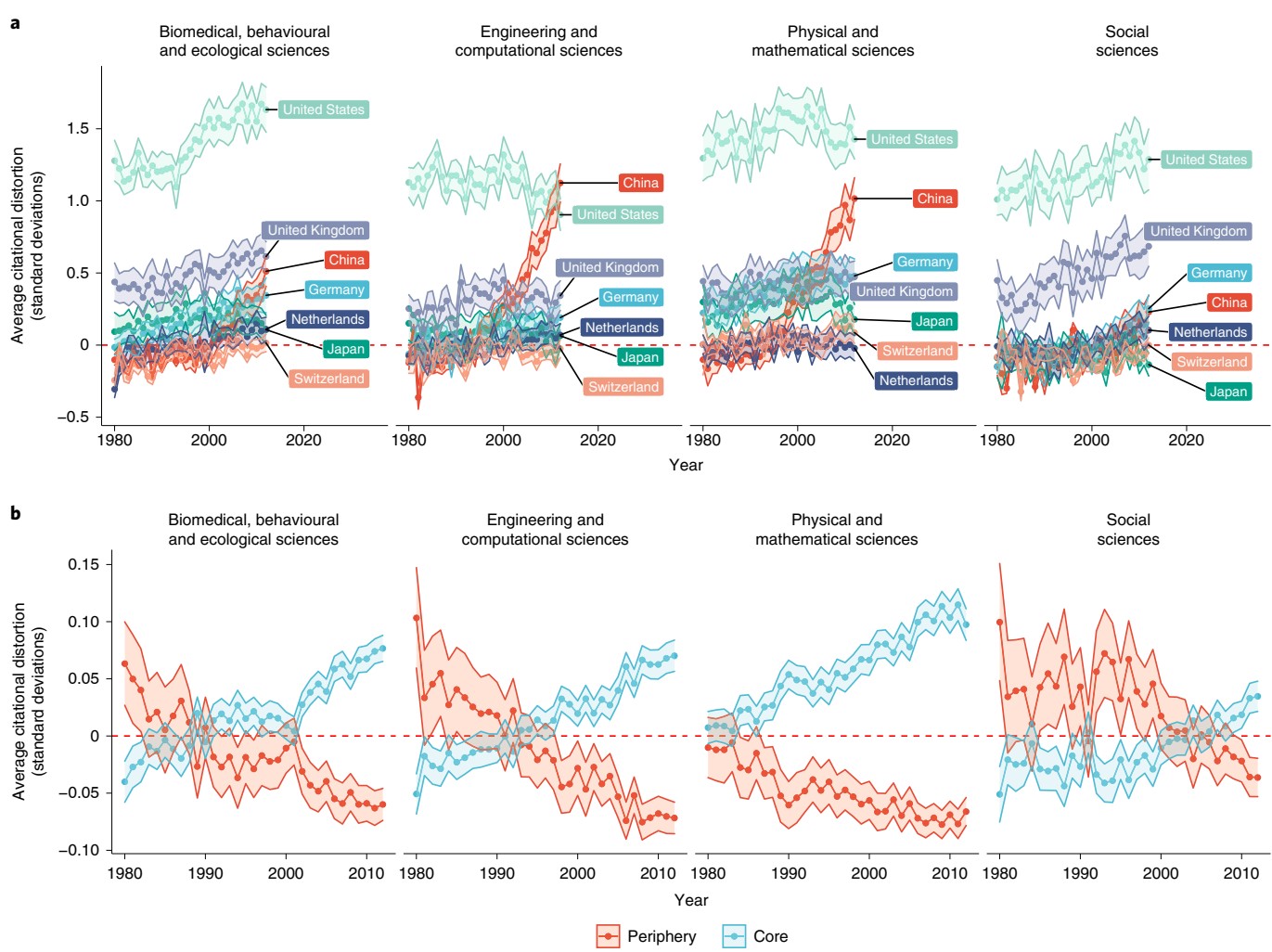

**Fig. 4 | Comparing global citational distortion by the type of field. a,b,** The trends plotted in Fig. 3, but parsed by the type of field. The shading around the trends denotes the standard errors of these averages. Note that citational distortion should be interpreted as the difference in the number of standard deviations between edges in the citation network (measured in terms of standard deviations relative to the edge weights in the citation network) and those in the text similarity network (measured in terms of standard deviations relative to the edge weights in the text similarity network).

Figure 6 plots the percentage of countries in the core and in the periphery that are being overcited or undercited in the year 2000 and in the year 2012 for different types of fields. Here the denominator is the number of countries present in the year 2000 or the year 2012, where summing the percentages vertically in each year results in 100%. The data points in the figure also contain the number of countries (given as *N*) and the average distortion value with its standard errors in parentheses.

Except for the physical and mathematical sciences, where the representation of countries in the overcited and undercited groups remained steady, the percentage of core countries that were overcited increased from 2000 to 2012. For the biomedical, behavioural and ecological sciences, the percentage increased from 9.68% of all countries in 2000 to 22.73% of all countries in 2012; for the engineering and computational sciences, from 13.64% of all countries to 22.73% of all countries; and for the social sciences, from 2.27% of all countries to 9.09% of all countries. Similarly, the percentage of periphery countries that were undercited rose over the same period. For the biomedical, behavioural and ecological sciences, the percentage increased from 40.32% of all countries in 2000 to 43.55% of all countries in 2012; for the engineering and computational sciences, from 39.83% of all countries to 45.76% of all countries; and for the social sciences, from 36.9% of all countries

to 42.86% of all countries. Notice as well that the changing representation of countries within the overcited and undercited groups does not necessarily correspond to the average distortion numbers within those groups.

Figure 7 uses the average distortion of each country in 2012 by field type to calculate transnational regional averages to include the four regions mentioned previously, as well as North America (the United States and Canada) and Oceania (notably including Australia and New Zealand). This figure was plotted in R using the package maps[22]. We created global maps based on the average distortion of countries within transnational regions in the year 2012 for the different types of fields. Figure 8 expands this map by plotting the average national distortion within each region and parsed by the type of field. This figure was also plotted using the package maps[22]. Note that for the transnational regional figures in Fig. 8a–d, the colour scales are mapped for the data of countries in those regions. Countries should not be compared in their colours across figures. The global map in Fig. 7 highlights the main regional differences for that purpose. There are several notable countries in each region, such as China in Fig. 8b and Brazil in Fig. 8c. In contrast, countries in Africa and the Middle East and most countries in South America are near parity, where they are cited to the same degree to which their research language aligns to other countries.

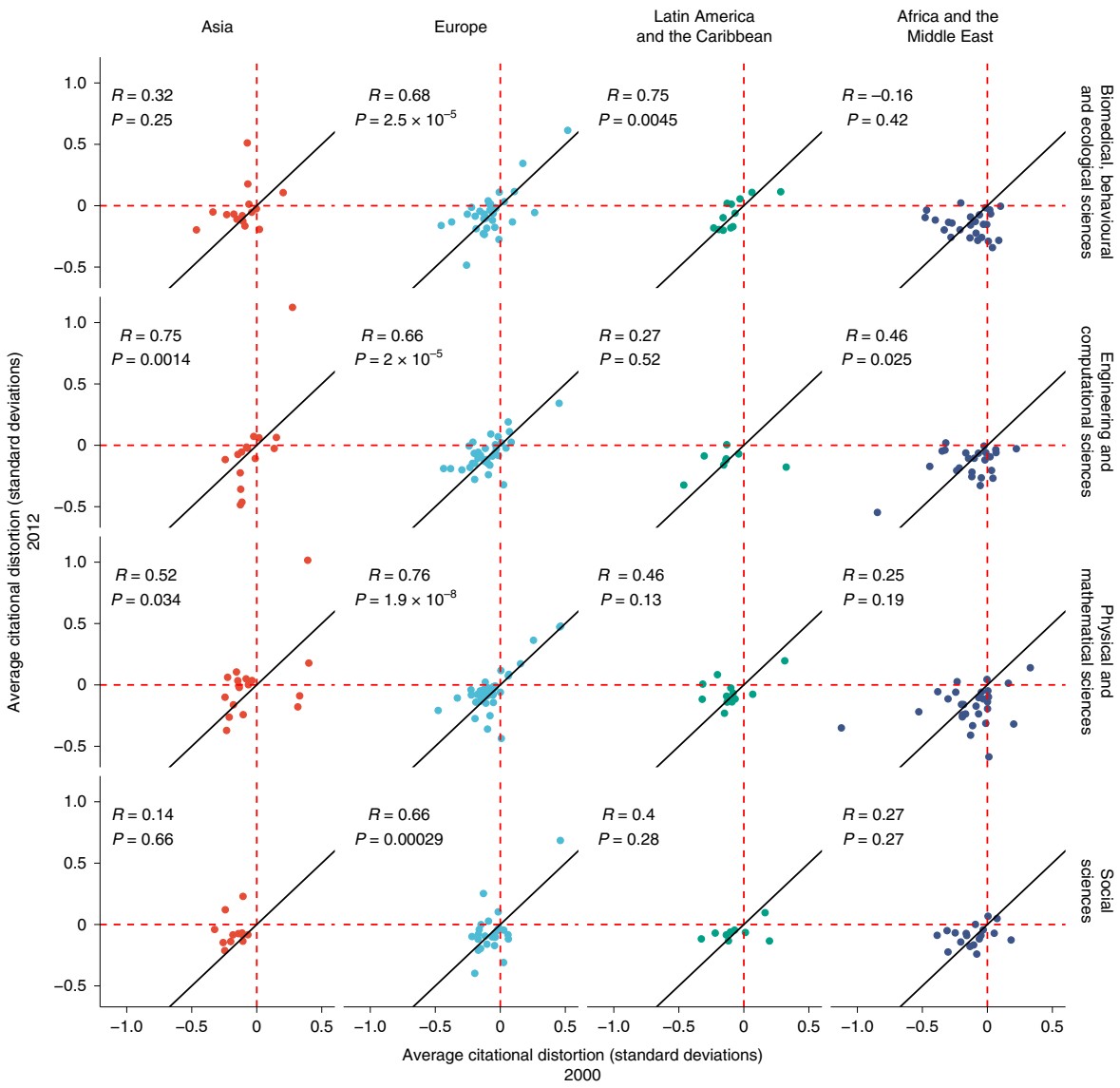

**Fig. 5 | Comparing citational distortion between 2000 and 2012.** The average distortion for countries in the year 2012, parsed by region and by the type of discipline. The trends for all countries in the years 2000 and 2012 are plotted on the *x* axis and *y* axis, respectively, but parsed by the type of field and by transnational region. R refers to the Pearson's product–moment correlation.

## Discussion

These results are especially interesting in light of the broader literature. The rise of China, for instance, has been noted with other bibliometric data[23–25], so for the country to also be experiencing an increase in citational distortion adds another element to this history. However, our results do not similarly re-affirm the temporal trends around Europe in recent work on citation inequality among elite researchers. Where that work showed that countries such as the Netherlands and Switzerland have a large and increasing share of elite researchers[5], Fig. 3a shows that while the Netherlands and Switzerland are increasingly reaping citations in excess of the textual similarity of their research, they fall far closer to the mean. Finally, it is noteworthy that the gap between countries with high levels and low levels of citational distortion is most pronounced in the physical sciences, considering that these fields are traditionally known for sharing the strongest sense of how to evaluate and integrate knowledge[26].

The main limitation of the citational lensing framework is one of measurement. Textual similarity between countries is an unavoidably noisy signal, and this affects the comparison with citations downstream. So, even though it is correct to say that the international inequalities revealed in our analyses are a matter of prominence and recognition, more precision will come only with future refinements to the methodology. We have been able to correct for citation inflation in Supplementary Figs. 10–15, and in Supplementary Figs. 16–21 we describe the results of a secondary analysis where we test our rudimentary controls for the quality of research by uncensoring journal selection. However, there are other factors that matter and could always be taken into account in the future.

These potential concerns are offset by a number of unique strengths that the citational lensing framework provides. One of the primary ones we have outlined is its adaptability. We have used nation-labelled LDA (NL-LDA) in tandem with the KLD to model the similarity of scientific text, but many other approaches could be used in their place to capture another nuance around language use in science. The entropy-based metrics advocated by Vilhena et al.[12] and Altmann et al.[13] would bring more attention to inefficiencies in international communication among scientists, to take one example.

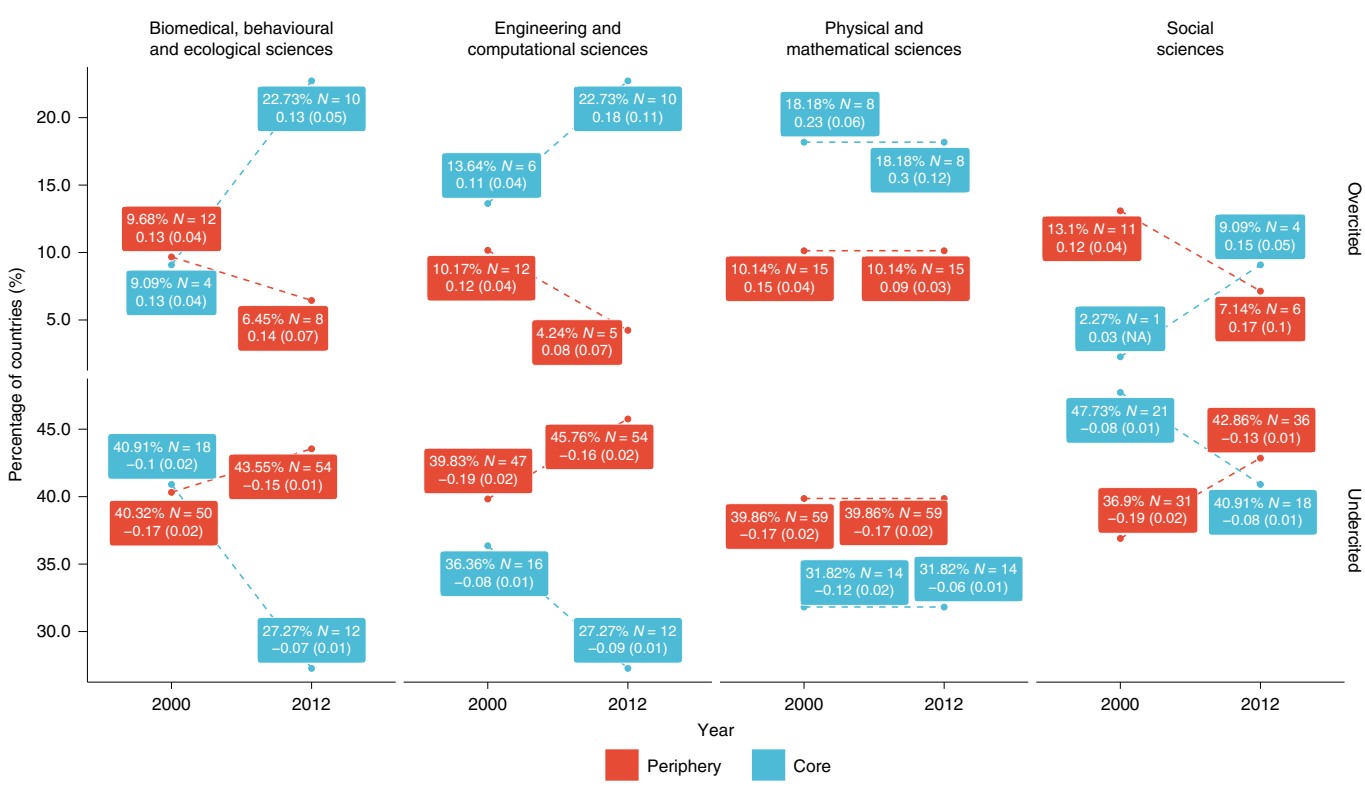

**Fig. 6 | Comparing citational distortion between core and periphery countries.** The percentage of countries in the core or periphery that are either overcited or undercited in the years 2000 and 2012, along with the average distortion for each group (standard errors are given in parentheses). Note: NA means not available.

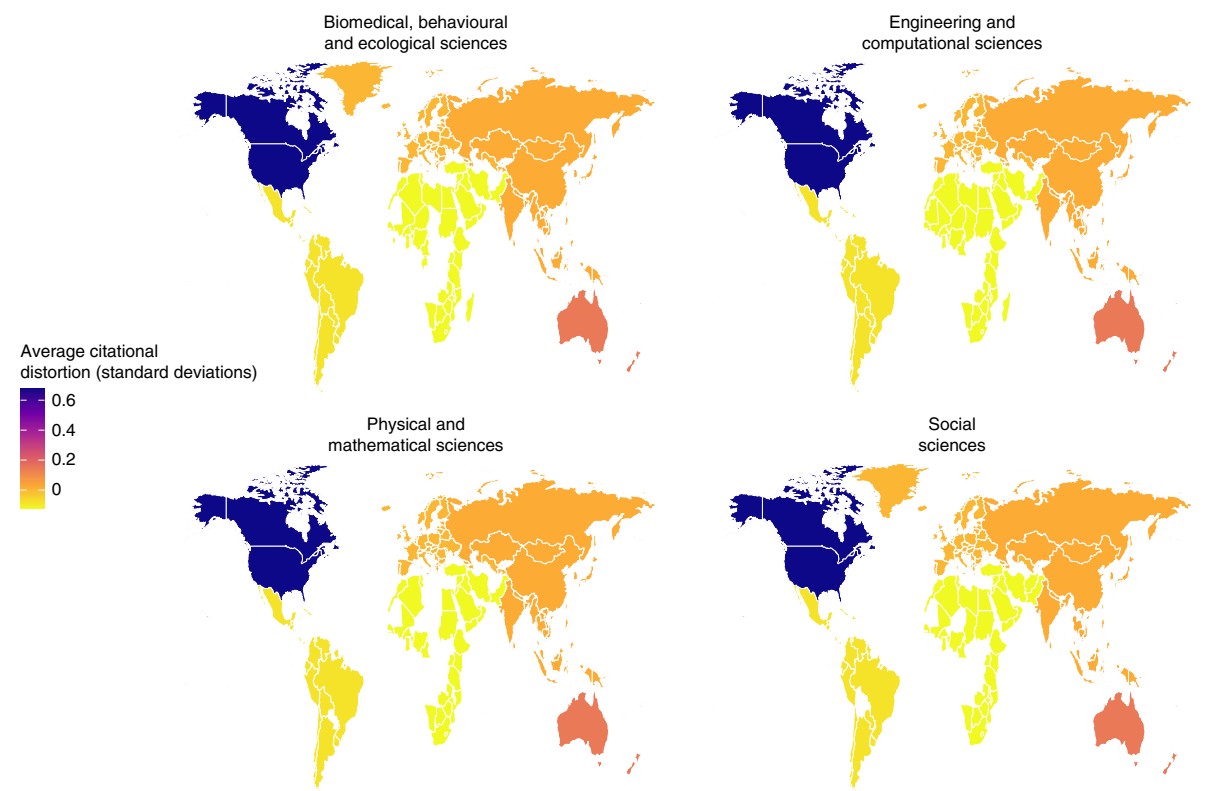

**Fig. 7 | Comparing citational distortion by transnational region in 2012.** The average distortion of countries mapped within transnational regions—Africa and the Middle East, Latin America and the Caribbean, Asia, North America, and Oceania—and by the type of field.

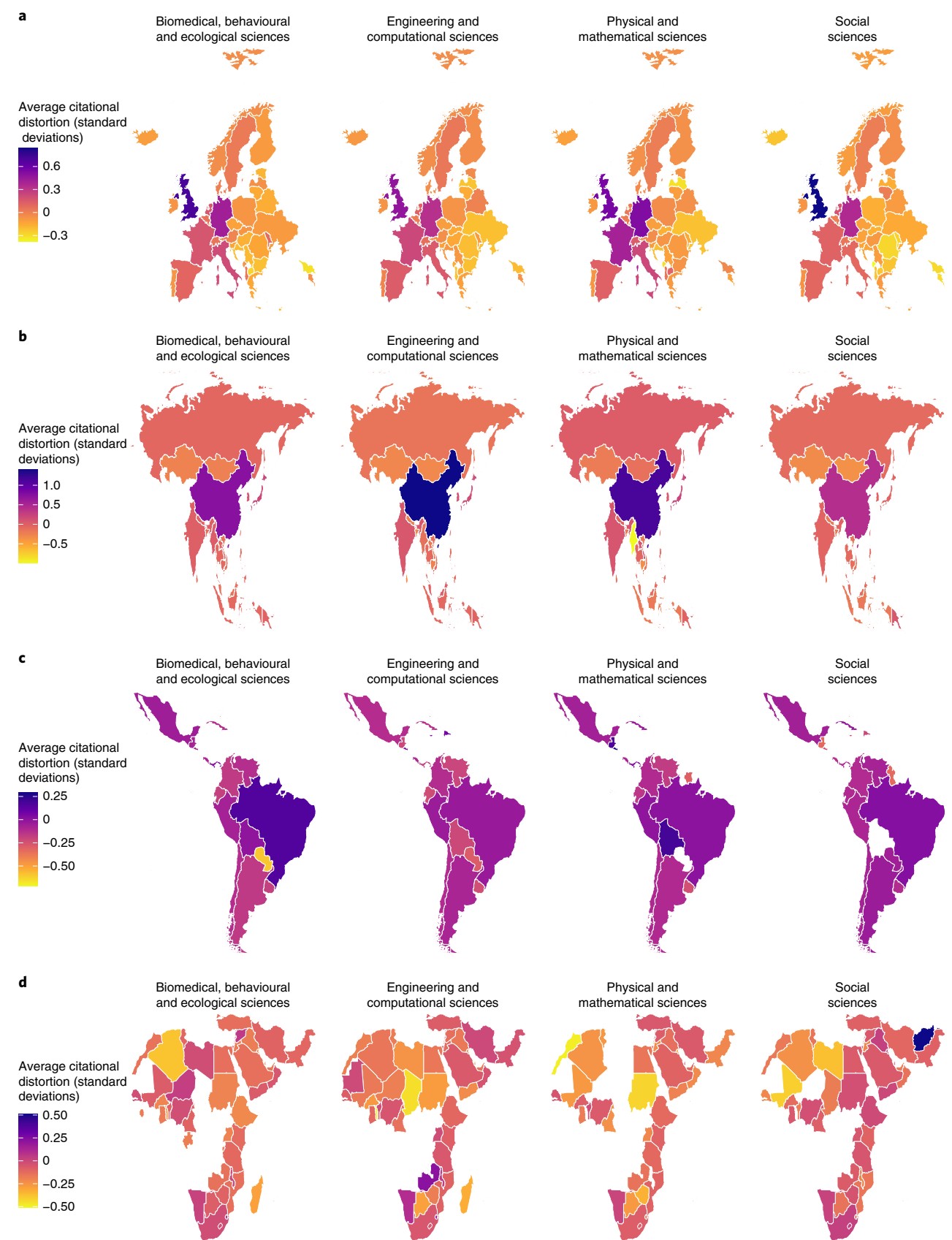

**Fig. 8 | Citational distortion within transnational regions in 2012. a–d**, The average distortion for individual countries in Europe (**a**), Asia (**b**), Latin America and the Caribbean (**c**), and Africa and the Middle East (**d**) in 2012. Note that the colour scale varies by region to highlight regional trends that are not visible when using a global colour scale.

The citational well may also be meaningful in contexts where nation-states are not the main feature of interest. Reputation effects among journals or universities, for instance, could be studied with the method we have described. The comparison may also be relevant to questions surrounding innovation and intellectual property, as patents are themselves embedded within a citation network and can be evaluated on the similarity of their language use. Any extension will have to be validated, but the simplicity of the method lends itself well to re-use in new contexts.

Regardless, identifying countries that do not receive citations in the amount we would expect given the subject matter of their research—whether they receive more citations or fewer—provides us with a tool to check on the efficiency of knowledge flow in science. Indeed, as demonstrated in Figs. 3 through 8, only a few highlighted countries are the real sole winners here, as the overall average $\beta$ coefficient trend plotted in Fig. 2b and the overall average citational distortion trend plotted in Fig. 3b are quite modest over 30 years. This holds real value for the study of scientific research. In the first place, citational lensing at the international level makes it possible to identify countries with a successful scientific enterprise. Citations should not be the sole arbiter for success in science, though the reality is that they continue to be disproportionately impactful as a metric. Countries that seem overcited by the approach we have taken here have been successful in this narrow sense. The empirical results bear out these points. Consider the United States and China: both countries reap far more citations than the language of their work would suggest. While this has been true for the United States for a long time, reflective of its large economy and substantial investment in scientific research, China's success in attracting citations has come more recently. More work would be needed to establish any causal connection, but the timing of China's rise in Figs. 3a, 4a and 8b is conspicuous given its major policy shift regarding science and technology in the 1990s.

This leaves citational lensing as a useful metric in tracking the effectiveness of national science policies, as well as evaluating the relative importance of various national factors in nurturing a highly cited scientific community. Otherwise, citational distortions will continue to impose limits on the circulation of knowledge, novel ideas and future innovations and are ultimately inefficient for sustained knowledge production. Better identifying who is undercited not only promotes the inclusion of often excluded perspectives but also enhances knowledge production. The exposure to diverse perspectives and abilities consistently improves outcomes in collective problem-solving ventures such as scientific research collaborations[27]. At the very least, overlooking research from wide swaths of the global scientific community means that knowledge remains unincorporated and human capital unused, especially in many rising middle-income countries with growing scientific enterprises.

Citational lensing also offers a way to study the differences in knowledge production between the Global South and the Global North, and the flow of knowledge between them. The long-standing concerns around the global inequalities in science have typically been substantiated with analyses of publication patterns or with rich qualitative interviews[2,28–31]. By bringing attention to the (mis)match between citations and textual similarity, citational lensing can help reveal more of the true impact that countries imprint on discourses across scientific fields and on the disproportionate attention and recognition that some research receives. Progress has been made by several countries in the semi-periphery towards this end, but each new entrant to the international scientific community continues to face a struggle in getting the appropriate recognition for the work of their scientists. That said, individual scientists probably derive benefits in their career trajectories from national reputations, though more research needs to be done to confirm this.

What remains to be seen is which factors do the most to distort citations from textual similarity. The visibility and quality of research are both likely contributors, alongside funding levels and overall reputations at the national level. However, as we do control for the growth in the number of journals over time, this may be one way to account for both the quality of published work and its subsequent visibility. Similarly unsettled is the issue of whether the distortion captured by the citational well could predict or stand in as a proxy for any of these same factors[32–34].

## Methods

To capture citational lensing, we represent science as a multiplex network, $\mathbf{L}$, with three layers (Fig. 1). Consider the simple case of citational lensing in a single field in a given year $t$. $\mathbf{L}_{\text{citation}}$ is the citation network between countries, where $\mathbf{L}_{\text{citation}_{i,j}}$ contains the citation flow from country $i$ to country $j$. To make things comparable across the layers of the multiplex network, $\mathbf{L}_{\text{citation}}$ is constructed as the number of citations received by country $i$'s papers published in the given field in year $t$ by all other countries $j$, where we use a five-year window after publication year $t$ to capture all citations from countries $j$ from year $t$ to year $t+5$. In that way, the text network based on published papers in year $t$ corresponds to the citation network of the number of cumulative citations received over the ensuing five years by papers published in year $t$. We use $z$-scores for the edge weights rather than the raw citation counts themselves.

Another layer, $\mathbf{L}_{\text{text}}$, is a network where each connection $\mathbf{L}_{\text{text}_{i,j}}$ is the similarity of the text of country $i$'s research output to that of country $j$. To capture the degree of similarity, we apply a unique supervised topic model called a labelled LDA model[9]. Using the nationalities of authors on papers, the NL-LDA model is unique in that it captures the extent to which ideas and concepts embodied by $n$-grams in the texts are associated to authors from which countries. This approach is useful to disentangle and establish what is being studied in different countries, as many papers are increasingly authored by researchers from different countries. The KLD[8] is taken for the similarity between countries in the text of their scientific papers. In our case, the KLD measures how much information is lost going from the text of one country $i$'s scientific output to that of another country $j$.

The reasoning here is similar to that used in other work in the science of science[12]. Information loss imitates the amount of work that scholars have to do to communicate their ideas. When very little information is lost, communication is seamless; when lots of information is lost, communication is difficult. Note that this is not a symmetrical relationship, and that is by design. The $\mathbf{L}_{\text{text}}^{\mathrm{T}}$ layer tends to identify the most common subject matter in national research, so when information is lost in moving from country $i$ to country $j$, it indicates that researchers in country $i$ publish about some topics that researchers in country $j$ do not (though this is, of course, usually a matter of degree rather than an issue of presence and absence). This means that it is harder on average for a scientist in country $i$ to find a counterpart in country $j$ that is working along a similar line of research than it is for a scientist in country $j$ to find a someone working on similar problems in country $i$. This also means that it is easier to find a paper from country $j$ that cites a paper from country $i$ than it is to find the reverse, assuming that citations are more likely when two papers have the same subject matter.

In principle, when the information loss is high going from $i$ to $j$, we say that the similarity of $i$ to $j$ is low. When very little information is lost going from $i$ to $j$, we say that the similarity of $i$ to $j$ is high. Just as in the citation layer, $z$-scores are used for edge weights, the only difference being that we take the negative here in the text layer, as high information loss implies exactly the opposite relationship that high citations imply. So, when we compare the multiplex layer $\mathbf{L}_{\text{citation}_{i,j}}$, which measures citational flow from country $i$ to country $j$, with $\mathbf{L}_{\text{text}_{i,j}}$, which captures how similar country $i$ is to country $j$, we use the transpose of $\mathbf{L}_{\text{text}}$ to result in $\mathbf{L}_{\text{text}}^{\mathrm{T}}$, where the similarity of country $j$ to country $i$ given as $\mathbf{L}_{\text{text}_{j,i}}^{\mathrm{T}}$ is equivalent to $\mathbf{L}_{\text{text}_{i,j}}$. $\mathbf{L}_{\text{text}}^{\mathrm{T}}$ is used in equation (1). We use this transpose because the more researchers in country $j$ cite researchers in country $i$, we posit that the work produced by researchers in country $i$ (that is, who is being cited and receiving the attention to the work being done in country $i$ with their citations) ought to be more similar to the work produced in country $i$ (that is, who is being cited and receiving the attention from country $j$). Distortions thus ought to reflect either over-recognition or under-recognition via attention (vis-à-vis citations) relative to the work being done elsewhere.

The third layer, $\mathbf{L}_{\text{distortion}}$, is what we call the citational well (drawing on the idea of gravity wells). This layer is constructed so that it will capture the difference between the other two layers, as given by equation (1) and Fig. 1. This means that every $\mathbf{L}_{\text{distortion}_{i,j}}$ represents the distortion in the citation flow from country $i$ to country $j$, relative to what we would expect on the basis of the similarity of the text written by country $i$'s scientists to that written by scientists in country $j$. Also implied is that the sum of the distortion for country $j$ relative to every other country in the network—country $j$'s in-degree in $\mathbf{L}_{\text{distortion}}$—represents the total distortion in the citation flow to country $j$.

To illustrate how citational lensing can be applied, we use nearly 20 million academic papers in nearly 150 fields and subfields from 1980 to 2012 in MAG, one of the most extensive metadata repositories of academic publications.

These data include metadata such as citations, along with the abstract text of published research articles. We show how citational lensing can be used to characterize changes in the international scientific hierarchy over time and how it can be scaled to cover all of science.

MAG classifies journals into various fields, which provides a fairly reliable reflection of disciplinary boundaries and allows for selection across a wide variety of fields. MAG uses a six-tiered field classification ID scheme that is human generated for the highest two levels. We primarily use the second-highest level, which offers these more granular field divisions. So instead of using just 'physics', we consider 'astrophysics' and 'nuclear physics' to be their own fields because they have different citation practices. Fields are identified and defined for our purposes as their field IDs in MAG, and the fields are itemized in Supplementary Table 1. We classify these fields into four broad categories: (1) biomedical, behavioural and ecological sciences; (2) engineering and computational sciences; (3) physical and mathematical sciences; and (4) social sciences. We use no other sort of field normalization. The population of journals in MAG increases considerably over time, which may partly affect the representation of countries in our analysis.

$L_{citation}$ is assembled using the citation data in MAG. As mentioned above, each $L_{citation_{ij}}$ holds the citation flow from country $i$ to country $j$. Because citation inflation[14] distorts the volume of cumulative citations in a field over time, rendering temporal comparisons biased, we standardize and 'deflate' the number of citations received in years $t+n$ to the equivalent number of citations that would have been received in the year $t$ that the paper was published. In essence, the citations received in a future $t+n$ year are converted into an exchange rate based on the year the paper was published $t$, rendering comparing citations across time less biased by volume. (In Supplementary Figs. 10–15, we rebuild our main figures comparing the citation deflation method that we use here to two other conditions: one that does not include any deflation and another that employs our own deflation method focusing specifically on countries.)

$L_{text}$ is constructed using text from the abstracts and titles of each paper. This has advantages over using the full texts of research papers, since some fields format papers to emphasize methods over theory or vice versa, and others might have a strict length criterion, in terms of word count or page length. Abstracts, however, succinctly summarize the most important concepts in a paper. We restrict our analysis to papers with English-only abstracts. (In Supplementary Figs. 22–27, we rebuild our main figures comparing these English-only abstracts to those that were subsequently translated from their original language into English by us using Google Translate.)

We build both $L_{citation_{ij}}$ and $L_{text}$ using only those journals that have existed in our data since 1980, the starting point of our analyses. (In Supplementary Figs. 16–21, we rebuild our main figures including all journals irrespective of their tenure in the data.) The important terms and phrases that represent ideas, concepts and phenomena need to be efficiently extracted from abstract texts. So, we construct each field's corpus in year $t$ as a combination of unigrams, bigrams and trigrams from every document's abstract, referred to as Field$_t$. For our analyses here, we use English-only abstracts to mitigate the risk of mistranslation. We also translate non-English abstracts using a Python module called googletrans that functions as an API with Google Translate and reconstruct our analyses, but our conclusions are consistent with what we present here. We apply a phrase extraction algorithm called RAKE (Rapid Automatic Keyword Extraction) to each abstract to extract all important phrases and terms from unigrams through trigrams[35]. RAKE extracts terms and phrases from abstracts by analysing the frequency of each $n$-gram and its co-occurrences with other $n$-grams in the text. An advantage of RAKE over other approaches is that it is domain independent, so it does not rely on a pretrained corpus to identify what terms are important. We then compiled an 'academic stop word' list of common phrases used in academic writing based on Coxhead[36] and removed them from the abstracts.

KLD compares probability distributions. To process the text of scientific articles so that each country has its own probability distribution, we apply NL-LDA models on abstracts from MAG publication abstracts to measure how similar or dissimilar the phenomena studied by researchers in different countries are[2–4]. We apply an NL-LDA model to each Field$_t$ corpus. This approach parses the influence of countries on multi-authored, international papers, a staple of many fields. We measure how similar individual countries' unique national signatures—or how strongly associated the terms found in a field's corpus in a year are associated to researchers in some country $x$—are to one another. The NL-LDA produces a matrix, $\varphi_{Field_t}$, where the rows are the $n$-grams in the corpus for Field$_t$ defined as $w_m$ and the columns are the national signatures defined as $C_n$. We standardize each national signature (column) in $\varphi_{Field_t}$ such that for each national signature, we assign zero values to all terms that were not present in papers authored from a particular country. (Our implementation of the NL-LDA model assigns a very small non-zero value to all terms that are not present in documents with a particular nation-label but are present in Field$_t$.) As the national signatures sum to 100%, we then renormalize each national signature after we convert the associative probabilities of absent terms to zero so that the national signature still sums to 100%.

We first validate the quality of the nation-labels produced by the NL-LDA using topic cohesion scores, the standard measure for how distinct a topic is from other topics derived from the same model. A cohesive topic forms a distinctive grouping

of its top $n$-grams that differentiates it from other topics. However, to date, no equivalent approach exists to measure nation-label cohesion for a supervised model like the NL-LDA in the same way as topic cohesion does for unsupervised models like the LDA. This is because the number of appropriate topics extracted from an LDA is variable and somewhat subjective, but the NL-LDA nation-labels are nominally fixed. That said, not every country may produce enough published papers in a year to produce meaningful results, so including every country in our analyses without any filtering may not be prudent. We apply the umass topic cohesion measure to the nation-labels in each NL-LDA model, where we compare the document co-occurrences of each nation-label's top 25 strongest associated terms from its national signature. Whereas with unsupervised LDA models, lower scores indicate more distinct and cohesive topics, with NL-LDA models, the opposite holds true: nation-labels with strong national signatures lead the way in global science and have lexical usage that is more widespread throughout the field. For each NL-LDA model, we convert these scores into percentile ranks, where the nation-labels that are the most ubiquitous (such as the United States and in later years China) are in the highest percentile (that is, they have lower coherence scores) and less active countries are in the lowest percentile (that is, they have higher coherence scores). For the results presented here, all of the nation-labels are included in the analyses. In Supplementary Figs. 1–7, we rerun nearly all of the figures presented here at the 25th and 75th percentiles. Our results broadly hold despite the exclusion of nation-labels.

With these matrices, we measure how similar any country's subject matter is to that of all other countries for some Field$_t$. However, a standard similarity score (like a cosine similarity) is not directed, and our aim is to understand how much one country looks like another where reciprocation may not happen. We compare every country to every other country in $\varphi_{Field_t}$ and take the KLD of every column in $\varphi_{Field_t}$ to every other column, where each comparison is a weighted, directed link that creates an international network of asymmetric text similarity. To calculate this score, we take the two vectors for a country $i$ and another country $j$, presented as their national signature vectors $c_i$ and $c_j$, respectively, to determine how similar they are to each other:

$$KLD(c_i \parallel c_j) = \sum c_i \log \frac{c_i}{c_j} \qquad (2)$$

Here KLD measures how much information is lost by national signature $c_i$ when approximated with the national signature from $c_j$. In other words, the less information that is lost by approximating $c_i$ with $c_j$, the more similar $c_j$ is to $c_i$. From here, we construct 4,914 international networks of topic similarity across nearly 150 academic fields and 33 years of data (that is, 1980 to 2012), defined as $KLD_{Field_t}$ (referred to in the results as $L_{text}^{T}$).

We create an upper bound for KLD in the following way: for each KLD network, $KLD_{Field_t}$, we take the negative of its $z$-score, so that the lowest value (that is, the lowest information loss and the most similar country dyad) is normalized to be the largest value relative to all other edge weights in the network (in terms of standard deviations). The dyad with the lowest raw KLD score is thus the dyad where the least amount of information is lost by approximating $c_i$ with $c_j$, so that country $i$ is highly aligned with country $j$. This approach is advantageous as it renders comparison across networks possible, particularly for extreme values.

**Statistics and reproducibility.** Our analyses were observational, and no statistical method was used to predetermine sample size.

**Reporting summary.** Further information on research design is available in the Nature Research Reporting Summary linked to this article.

## Data availability
All data that were used to create the figures are available on the Harvard Dataverse at https://doi.org/10.7910/DVN/WCOINR.

## Code availability
All code that was used to perform the analyses and to construct the figures is available on the Harvard Dataverse at https://doi.org/10.7910/DVN/WCOINR.

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

## Acknowledgements

We thank D. McFarland, W. Powell, A. Goldberg, J. P. Alperin, B. L. Boerjesson, B. Keep, L. Rodman, S. Muñoz-Najar Galvez, J. Sohn, E. Mäkinen, E. Evans, the UCLA workshop on computational social science participants and the CUNY Faculty Fellowship Publication Program spring 2020 participants (K. Chen, M. Dibello, N. Hougaard, A. Lambert, E. Minei and M. Sanchez.), who all provided insightful and thoughtful comments on various versions of this article. This work is supported by funding from the NSF (award no. 2022395): 'Inequality in Global Scientific Research: Implications for Novelty and Innovation' PI: (CUNY-QC), C.J.G. Any opinions, findings, and conclusions or recommendations expressed in this material are those of the author(s) and do not necessarily reflect the views of the NSF. The funders had no role in study design, data collection and analysis, decision to publish or preparation of the manuscript.

## Author contributions

C.J.G. conceptualized the initial project and its methodology, performed the analyses, coordinated the project, and drafted, reviewed and edited the paper. A.C.H. contributed to conceptually developing the project and its methodology and drafted, reviewed and edited the paper. P.P. contributed to conceptually developing the project and its methodology and reviewed the paper.

## Competing interests

The authors declare no competing interests.

## Additional information

**Correspondence and requests for materials** should be addressed to Charles J. Gomez.

# Reporting Summary

Nature Research wishes to improve the reproducibility of the work that we publish. This form provides structure for consistency and transparency in reporting. For further information on Nature Research policies, see our Editorial Policies and the Editorial Policy Checklist.

## Statistics

For all statistical analyses, confirm that the following items are present in the figure legend, table legend, main text, or Methods section.

| n/a | Confirmed | |
|---|---|---|
| ☐ | ☒ | The exact sample size ($n$) for each experimental group/condition, given as a discrete number and unit of measurement |
| ☒ | ☐ | A statement on whether measurements were taken from distinct samples or whether the same sample was measured repeatedly |
| ☐ | ☒ | The statistical test(s) used AND whether they are one- or two-sided *Only common tests should be described solely by name; describe more complex techniques in the Methods section.* |
| ☐ | ☒ | A description of all covariates tested |
| ☐ | ☒ | A description of any assumptions or corrections, such as tests of normality and adjustment for multiple comparisons |
| ☐ | ☒ | A full description of the statistical parameters including central tendency (e.g. means) or other basic estimates (e.g. regression coefficient) AND variation (e.g. standard deviation) or associated estimates of uncertainty (e.g. confidence intervals) |
| ☐ | ☒ | For null hypothesis testing, the test statistic (e.g. $F$, $t$, $r$) with confidence intervals, effect sizes, degrees of freedom and $P$ value noted *Give P values as exact values whenever suitable.* |
| ☒ | ☐ | For Bayesian analysis, information on the choice of priors and Markov chain Monte Carlo settings |
| ☒ | ☐ | For hierarchical and complex designs, identification of the appropriate level for tests and full reporting of outcomes |
| ☐ | ☒ | Estimates of effect sizes (e.g. Cohen's $d$, Pearson's $r$), indicating how they were calculated |

*Our web collection on statistics for biologists contains articles on many of the points above.*

## Software and code

Policy information about availability of computer code

| Data collection | Data are primarily from the publicly available Microsoft Academic Graph (MAG). |
|---|---|
| Data analysis | Data were analyzed, processed, and visualized using Python 3, R, and SQL. The data are stored on Athena (Amazon), queried and read in to Python, analyzed with Python, and visualized with R. |

For manuscripts utilizing custom algorithms or software that are central to the research but not yet described in published literature, software must be made available to editors and reviewers. We strongly encourage code deposition in a community repository (e.g. GitHub). See the Nature Research guidelines for submitting code & software for further information.

## Data

Policy information about availability of data

All manuscripts must include a data availability statement. This statement should provide the following information, where applicable:
- Accession codes, unique identifiers, or web links for publicly available datasets
- A list of figures that have associated raw data
- A description of any restrictions on data availability

All data derived from the Microsoft Academic Graph (e.g., corpora, NL-LDA models, CSV edgelists, etc.), along with all relevant code in Python 3 and R used to analyze and visualize these data, will be made available at the Harvard Dataverse upon publication.

# Field-specific reporting

Please select the one below that is the best fit for your research. If you are not sure, read the appropriate sections before making your selection.

☐ Life sciences ☒ Behavioural & social sciences ☐ Ecological, evolutionary & environmental sciences

For a reference copy of the document with all sections, see nature.com/documents/nr-reporting-summary-flat.pdf

# Behavioural & social sciences study design

All studies must disclose on these points even when the disclosure is negative.

| | |
|---|---|
| Study description | The data are quantitative and text-based. The text data were processed using nation-labeled LDA models and using MRQAP network regression models. |
| Research sample | We selected 181 field IDs from the Microsoft Academic Graph (MAG) and read in all of the published paper metadata from papers published between 1980 and 2015, inclusive. |
| Sampling strategy | No sample size calculations were performed. Fields in MAG are classified into a several-tier hierarchy that progresses deeper into field sub-specialties. We selected fields from tier-1 which has the most representative set of fields without selecting from sub-specialties. |
| Data collection | Data were procured from the publicly available Microsoft Academic Graph (MAG). |
| Timing | Our metadata in the Microsoft Academic Graph (MAG) extend to 2017. We stop our analyses in 2015 since the total number of papers in 2017 drops compared to 2016, indicating that these data may not be fully complete. |
| Data exclusions | Only papers that had abstract text data were used in our analyses. |
| Non-participation | Participants were not used in this study. |
| Randomization | Randomization strategies were not relevant to this study. |

# Reporting for specific materials, systems and methods

We require information from authors about some types of materials, experimental systems and methods used in many studies. Here, indicate whether each material, system or method listed is relevant to your study. If you are not sure if a list item applies to your research, read the appropriate section before selecting a response.

## Materials & experimental systems

| n/a | Involved in the study |
|---|---|
| ☒ ☐ | Antibodies |
| ☒ ☐ | Eukaryotic cell lines |
| ☒ ☐ | Palaeontology and archaeology |
| ☒ ☐ | Animals and other organisms |
| ☒ ☐ | Human research participants |
| ☒ ☐ | Clinical data |
| ☒ ☐ | Dual use research of concern |

## Methods

| n/a | Involved in the study |
|---|---|
| ☒ ☐ | ChIP-seq |
| ☒ ☐ | Flow cytometry |
| ☒ ☐ | MRI-based neuroimaging |

