## [Peer Review File · Nature Human Behaviour]

Peer Review Information

Journal: Nature Human Behaviour

Manuscript Title: Leading countries in global science increasingly receive more citations despite doing similar research as less cited countries

Corresponding author name(s): Charles J. Gomez

Reviewer Comments & Decisions:

Decision Letter, initial version:

30th March 2021

Dear Dr Gomez,

Thank you once again for your manuscript, entitled "Rising Global Inequality in the Acknowledgement of Scientific Research.", and for your patience during the peer review process.

Your Article has now been evaluated by 3 referees. You will see from their comments copied below that, although they find your work of potential interest, they have raised quite substantial concerns. In light of these comments, we cannot accept the manuscript for publication, but would be interested in considering a revised version if you are willing and able to fully address reviewer and editorial concerns.

We hope you will find the referees' comments useful as you decide how to proceed. If you wish to submit a substantially revised manuscript, please bear in mind that we will be reluctant to approach the referees again in the absence of major revisions. We are committed to providing a fair and constructive peer-review process. Do not hesitate to contact us if there are specific requests from the reviewers that you believe are technically impossible or unlikely to yield a meaningful outcome.

To guide the scope of the revisions, the editors discuss the referee reports in detail within the team, including with the chief editor, with a view to (1) identifying key priorities that should be addressed in revision and (2) overruling referee requests that are deemed beyond the scope of the current study. We hope that you will find the following summary to be useful when revising your study. Please do not hesitate to get in touch if you would like to discuss these issues further.

In their reviews of your manuscript, Reviewers 2 and 3 raise fundamental concerns about measurement validity and the extent to which your measure actually captures country-level citation biases. These concerns are crucial from an editorial perspective. Country-level citation bias is currently

confounded with numerous other potential sources of citation distortion and in revision we expect extensive additional analyses that de-confound the effect of country-level citation bias from other sources of citation distortion, as well as a more careful characterization of what your measure actually captures. We also expect that all other technical and conceptual points raised by the reviewers will be carefully addressed.

If you wish to submit a suitably revised manuscript we would hope to receive it within 6 months. We understand that the COVID-19 pandemic is causing significant disruptions which may prevent you from carrying out the additional work required for resubmission of your manuscript within this timeframe. If you are unable to submit your revised manuscript within 6 months, please let us know. We will be happy to extend the submission date to enable you to complete your work on the revision.

- Include a "Response to the editors and reviewers" document detailing, point-by-point, how you addressed each editor and referee comment. If no action was taken to address a point, you must provide a compelling argument. This response will be used by the editors to evaluate your revision and sent back to the reviewers along with the revised manuscript.
- Highlight all changes made to your manuscript or provide us with a version that tracks changes.

[REDACTED]

Thank you for the opportunity to review your work. Please do not hesitate to contact me if you have any questions or would like to discuss the required revisions further.

Sincerely,

Charlotte Payne
Editor
Nature Human Behaviour

Reviewer expertise:

Reviewer #1: Global citation inequality; Sociology of science

Reviewer #2: Global citation inequality; Sociology of science

Reviewer #3: Global citation inequality; Network science, complex networks

REVIEWER COMMENTS:

Reviewer #1:

Remarks to the Author:

This is a peer review of the paper titled "Rising Global Inequality in the Acknowledgement of Scientific Research" by Gomez, Herman and Parigi, submitted to Nature Human Behaviour.

The manuscript describes a framework which combines a citation network with a textual similarity network into an advanced "citational well", which allows the authors to analyze the difference between the citations a country receives on a given topic versus what would be expected from the topics covered by this country. This allows for a highly interesting analysis of the inequality of citation distributions between countries, hinting to a country-level Matthew-effect, but also opens up for a discussion of knowledge flows and the success of different countries. The presented method has tremendous potential for further research as well.

This is a highly original and novel study, and it presents some of the most brilliant work I have seen in this field for a long time. I very highly recommend it for publication, but have some comments which I would like the authors to consider.

1. It is well known that e.g. the Web of Science has a "database effect" when studying temporal trends. The inclusion of additional sources over time and especially the increased output of countries like China can have large consequences for the annual citation distributions, and I strongly assume these effects extend to the citation networks used in this study. Have the authors looked into whether there are temporal database effects similar to this in Microsoft Academic Graph? And if so, what are the consequences for their analysis?

I appreciate that the authors split the citation network into field-specific networks. However, it is not completely obvious to me how these fields are defined, or whether field differences in referencing norms and citation intensity could perhaps give greater weight to one field over another in the overall analyses. It is also my impression that the field definitions are too broad to capture differences in these norms which for instance vary immensely between physics and mathematics, and even within physics between experimental and theoretical physics. I would appreciate a discussion of how this affects the analysis, and if perhaps a more fine-grained standardization might be necessary.

It seems like a good call to avoid cosine similarity in the text similarity measurements. Not only is it undirected, as noted by the authors, but it can also be difficult to discriminate between units with this measurement. But I wonder whether the authors have tested the discriminatory power of their chosen metrics? If not, a test of this could be very useful for informing us on the robustness of the text-similarity layer.

Reviewer #2:

Remarks to the Author:

Thank you for giving me the opportunity to review this interesting paper, which introduces a new method for gauging the correspondence between citation flows and textual similarity across scientific nations. The proposed method is innovative and the paper has some clear qualities and contributions. However, I do not think that it is suitable for publication in Nature Human Behaviour in its current form.

If the authors are invited to resubmit the paper, I would be very curious to hear their responses to the following questions.

1. Is this method designed to estimate country-related citation bias? Although the language is a bit ambiguous on this matter, it seems that the authors think of their approach as a method for measuring the level of citation bias in favor of core scientific nations. Specifically, they refer to the "uncomfortable spaces" in science, where citation networks and textual similarity are not aligned and to the "citational distortion" that will "continue to impose limits on the circulation of knowledge, novel ideas and future innovations". They also talk about countries as being over-recognized and under-recognized. While I commend the authors for putting the important question of global citation disparities on the scholarly agenda, I do not think their method will allow us to make any claims regarding country-level citation biases. I strongly encourage the authors to be clearer and more explicit about what they can and cannot conclude based on this approach. At its core this is a question of measurement validity. I miss a clearer statement on what it is this method actually measures and what it does not measure.
2. Is this a methods paper or an empirical paper? In the discussion section, the authors frame their paper as both a methods contribution and an empirical contribution. The paper would probably be strengthened if one of these two types of contributions was given priority throughout the paper. If this is a methods contribution, I think it would be appropriate to include a more comprehensive validation of the proposed method's performance compared to alternative approaches. If this is primarily an empirical contribution aimed at displaying global disparities, I would like to see more considerations on how to deal with database effects, variations in citation practices across sub-fields (that are distributed unequally across nations), the potential sensitivity of the labelled LDA and the relative entropy score to enormous variations in sample sizes (N papers per country).
3. Why did the authors include non-English abstracts? In the Methods section, the authors describe that they've used a python package to translate non-English abstracts into English. I'm wondering how this may affect the validity of their findings? Is it reasonable to expect that an a German or Italian research would cite a highly (textually) similar paper in Portugese compared to a highly similar paper in English? I don't think so. To what extent may the estimated "citation distortion" be driven by language differences. It would be great to see the same analysis done without non-English abstracts to answer this question.

4. To what extent are the main results in Figure 2 driven by database effects (including the increasing representation of countries, low-impact journals, and non-English journals in MAG over time). Figure 2a, shows a notable increase in the number of countries that are present in both the international text similarity network and the international citation network over time. Figure 2b and 2c are used to demonstrate a widening gap between core scientific nations and the “all-inclusive” country sample in terms of recognition obtained, given level of textual similarity. In this sense, these figures compare the core sample to itself over time (it’s both part of core and all-inclusive). As shown in the figures, the gap between the core sample and the full sample is much smaller in 1980 than in 2015, which makes sense, since the core-sample and the all-inclusive sample are far more likely to overlap in 1980 than they are in 2015. So is inequality really increasing, or does Figure 2B and 2C merely reflect the increasing number of non-core countries with lower-impact papers in the database over time? It would be interesting to see this analysis redone with two non-overlapping samples, core vs. non-core. The population of journals in MAG may be expected to increase considerably over time, which may partly drive the increase in country representation displayed in Figure 2A. Like Web of Science, MAG probably includes a smaller set of core (high-impact) journals in the early phase (1980), with an increasing number of peripheral (low-impact) and non-English journals included over time. This begs the question as to whether the trends observed in Figures 2b and 2c are driven by language effects (non-English papers citing English papers but not the other way around) and journal effects (a higher representation of core countries publishing in high impact journals?). It would be interesting to examine how the following adjustments may affect these trends: 1) removing non-English papers from the analysis. 2) focusing on the same “core-set” of journals for all years from 1980 to 2015.

5. To what extent may the increasing gap observed in Figure 2B and C be driven by citation inflation. According to the citation inflation principle (<https://www.sciencedirect.com/science/article/pii/S0048733319301003>), the average number of citations per paper will increase over time (as a result of a growing population of citing papers). This begs the question as to whether the increasing gaps observed in figure 2b and 2C are driven by citation inflation as opposed to an actual increase in the acknowledgement gap (same for Fig 4). It would be interesting to see if the results are similar, if the authors adjust for citation inflation.

6. According to the literature, it appears that the relative entropy similarity score is insensitive to sample size (number of papers per country per year). However, I’m wondering if there’s any way for the authors to verify this underlying assumption in their own study where labelled LDA and the entropy similarity score are combined. Looking at Figure 3A it seems that the huge research economies are scoring highest on the text-similarity measure and that China’s upwards trend corresponds with the rapid expansion of the country’s research system since the 1990s. This made me think of a possible bias, where larger research nations (with more papers per year) will score higher on the similarity measure given their size alone? It would be great if the authors could reflect a bit more on the question of sample sizes in relation to their similarity score.

7. On page 11 (bottom), the authors state that prominent countries like the United states, UK, Germany and China are the most central in both the centrality and citation networks. However, what about countries like Switzerland and the Netherlands, which are known to be the top global performers in terms of average citation impact per paper? On p. 11 it is also stated that “the US has an average Z-score of over 7 in 2017, meaning that the U.S. receives over 7 standard deviations more incoming citations above the mean number of incoming citations of countries across fields in 2017. However,

how much larger is the US in 2017 compared to the average country in terms of papers published per year? I think reflecting more explicitly on scale in these accounts would be appropriate. Likewise, it seems relevant to emphasize that biomedical research (with a high publication output) may be proportionally larger in the U.S. than in the "average" country used as a baseline, which may skew these results (due to lack of field-normalization).

Smaller issues:

In the abstract, it is stated that "We apply a new framework called citational lensing to identify where citations should appear between countries but are nonetheless absent given that what is embedded in their published texts is highly similar." Please consider revising "published texts" to avoid giving the impression that these analyses were made on full text material.

The following sentence is missing an "around" after the "center": "We find that scientific communities increasingly center research from highly active countries while overlooking work from peripheral countries"

Throughout the manuscript, the authors refer to a list of "core scientific nations" (including Western Europe, Australia, New Zealand, Eastern Asia and North America). How did they authors come up with this list of core-scientific nations. And in which ways can they be considered core nations? And please list all of these countries.

Reviewer #3:

Remarks to the Author:

Review for "Rising Global Inequality in the Acknowledgement of Scientific Research" by C. J. Gomez, A. C. Herman, P. Parigi

The paper "Rising Global Inequality in the Acknowledgement of Scientific Research" proposes a new framework to evaluate a relevant phenomenon in the science of science: the inequality in the number of scientific citations accruing to papers over time.

The authors' proposal rests on the assumption that the motivations for inequality in citation patterns are mostly country related (even though not clearly identified in the text) and hence one should compare the number of citations to one country from the others against the likely level of acceptance that a country would have obtained if distortion in citation patterns would be absent.

For this reason, the authors use a LDA model trained on paper abstracts to identify a measure of country specificity in the overall science output. Their data come from nearly 20 million papers across nearly 40 years and 181 fields from the Microsoft Academic Graph. The authors compare each country "fingerprint" to the ones of other countries obtaining a similarity network for the science output and hence a proxy of how science from a certain country could be easily understood (and thus cited) by the others. Comparing the similarity network with the citation network, they obtain a measure of distortion which they decompose by scientific area and geographic macroareas.

There are several flaws in the paper, which show a lack of care in presenting the results of the research. I am listing these problems in a separate section below.

However, there is a major flaw in the research design that in my opinion renders the paper not suitable for publication.

Major point

It is clear that citation patterns reflect a distorted scientific landscape, where certain scientific products receive a disproportionate attention with respect to others. Moreover, it is undoubtable that historical and country-specific reasons can be identified as relevant factors which have affected citation distortion to a large extent. However, starting from this evidence the paper makes three assumptions which are implausible:

1. There exists a common profile across all papers from the same country (or field and country couple) which can be inferred.
2. That profile can be obtained by solely looking at the way papers are written which can be approximated by their abstract.
3. The profile represents a credible representation of the actual value of the scientific output of a country (or field and country couple) which can be used to evaluate distortion in scientific paper success.

While the second point is more technical, the reason why the two other assumptions are unlikely is intuitive: they do not account for the way the scientific process works. To infer distortion in the citation process, a paper should be compared with other papers regarding the same topic in the same field. Furthermore, its position in the network of citations should be accounted. Then, assuming that the paper results are as relevant as those of other comparable papers, distortions may arise from several sources. Citation inflation in a specific field or period is one of them, country bias, as claimed by Gomez et al, is another. However, country-dependence must be distinguished by the effect of the co-authorship network in which the paper authors are embedded and by the centrality of the authors' institutions in the network of academic institutions. It is after accounting for all these measurable confounding factors (which, by the way, are very common in the literature), that other unobservable country-specific features may be addressed, for instance with an LDA model.

The authors not only ignore all these alternative explanations but use a process of construction of their dependent variable (the similarity network obtained via LDA) which artificially collapse all the differences on the country dimension, hence selecting ex-ante what results they will observe.

For this reason, I do not recommend this paper for publication.

Minor points

- The paper is badly written. There are several passages which are hard to understand and need to be rewritten in a clearer manner. Oftentimes colloquial English is used. There are several typos.
- The literature review is approximate: many passages, especially the more crucial ones, are not sustained by relevant references. Moreover, sometimes the authors' claims are vague and too general.
- The plots are not appropriate. First, they are not well explained in the captions. Some (non-exhaustive) examples: annotations in Figure 1 are completely ignored in the caption; the caption in Figure 2 does not say that Figure 2A represents violin plots, does not explain what is "All" and what is "Core", does not explain what are the vertical lines in all the plot. Furthermore, the plots sometimes are unnecessarily complicated (what is the reason for the noisy black area in Figure 4A and 4C?) and sometimes repeat useless details (differences among countries and fields are very hard to grasp in Figure 3).
- The multiplex modeling is not useful for the analysis or the discussion of results and the relation between the similarity and the citation networks could be explained without introducing the complexity of a multiplex representation.

Author Rebuttal to Initial comments

The reviewers' comments are in normal text, **our responses are in bold and indented**, and any changes to the manuscript's texts are bordered.

Reviewer 1

It is well known that e.g., the Web of Science has a "database effect" when studying temporal trends. The inclusion of additional sources over time and especially the increased output of countries like China can have large consequences for the annual citation distributions, and I strongly assume these effects extend to the citation networks used in this study. Have the authors looked into whether there are temporal database effects similar to this in Microsoft Academic Graph? And if so, what are the consequences for their analysis?

We thank Reviewer 1 for this excellent observation. Indeed, database effects are a serious cause for concern for any temporal analyses, in particular when attempting to best address the dramatic rise of Chinese-authored papers.

We have revamped our analyses, streamlined our results, and performed several robustness checks specifically targeted towards accounting for these issues.

First, we now construct our citation network as the number of citations received by papers published by country x in Fieldt within a five-year time window after its publication, whereas we previously just counted the number of citations received in each year. This more directly offers a 1-to-1 mapping of what is being cited to what is being researched in a given year. As such, our analyses now conclude in the year 2012 to account for all of the citations received in the following five-year window from 2012 to 2017.

Second, and following from astute observations from the other reviewers, we have taken the following steps to address these database effects. These efforts are documented in the Supplemental Materials and in this memo. In short, we re-ran our analyses and reconstructed our main figures to look at the impact of (1) English-only versus English and translated into English abstracts, (2) citation deflation techniques, (3) a broadened MRQAP analysis, (4) censoring the population of journals in our data and (5) hierarchical linear models (HLMs) of the number of papers per country.

A potential source of artificial growth captured by these database effects is the number of journals that are included in MAG. We re-run most of our analyses where both citations and our corpora are censored to journals that exist every year in our data, between 1980 and 2012. These data are now included in the main text instead of the uncensored version we used previously. In the Supplemental Materials, we also compare the journal censored versus the non-journal censored results and again find that our findings still hold.

Another bias may be the inclusion of abstracts translated into English when constructing the text similarity network, as the inclusion of these journals may be skewed towards more recent years and potentially distort the growth seen in many of the figures. We have restricted our abstracts used to build our NL-LDA models and citation networks to English-only abstracts. In the Supplemental Materials, we re-run most of our analyses comparing the English-only results with those that include both English abstracts and those translated into English using Google Translate.

A source of bias is the artificial growth of citations due to growing volume of papers. We have implemented two versions of citation deflation based on Petersen, et al.'s work on citation inflation: (1) the field-level measure from Petersen, et al. and (2) a country-field-level measure that extends Petersen, et al.'s field-level approach. On the latter, we calculate the total number of papers that Country i has at least one authorship on for each Field t . Much like Petersen et al.'s ratio construction for the total number of papers in the field, we instead deflate at the dyadic level based on the receiver's number of papers in year t to a baseline year. So, if we are looking at the number of citations that the United States received from China, we deflate this number based on the receiver (i.e., the United States) and its number of papers in year t to the baseline year. After all the dyads are deflated, we then calculate the indegree centralities for each receiver using the deflated edge weights. For both field-level and our country-field-level deflation measures, we pick the year 2000 as our baseline year that we choose to both inflate (prior to 2000) and deflate (after 2000) citations. Here, we compare these three treatments (field deflated, country deflated, and no deflation). Our findings still hold despite these deflation metrics.

I appreciate that the authors split the citation network into field-specific networks. However, it is not completely obvious to me how these fields are defined, or whether field differences in referencing norms and citation intensity could perhaps give greater weight to one field over another in the overall analyses. It is also my impression that the field definitions are too broad to capture differences in these norms which for instance vary immensely between physics and mathematics, and even within physics between experimental and theoretical physics. I would

appreciate a discussion of how this affects the analysis, and if perhaps a more fine-grained standardization might be necessary.

This is a great comment, and we again thank the reviewer for it. We agree that any form of aggregation across fields will raise questions around how best to adjust for differences between fields. Like we mention above, this is primarily an issue of how the results are aggregated, as opposed to being fundamental to the method we put forward.

However, this is a fair question when it comes to the empirical demonstrations, especially for the aggregated nation-level data. Countries are compared on similar terms, though some error is certainly introduced when, within a given field label, countries are differentially involved in some subfields rather than others (e.g., experimental versus theoretical physics).

MAG uses a six-tiered field classification ID scheme that is human generated for the highest two levels. To your point, since the first tier uses roughly twenty (or so) broad classifications like biology and physics, we use the second highest level that offers these more granular field divisions, like astrophysics and nuclear physics that have different citation practices. (These fields are listed in the Supplemental Materials.) Indeed, anything beyond the second tier is generated using clustering techniques and a “sub-field” label is assigned afterward. As such, going beyond our chosen field-level and disaggregating them further likely will not yield field boundaries that are less consistent than the human generated ones.

In the results, we included a short discussion and rationale behind this decision.

It seems like a good call to avoid cosine similarity in the text similarity measurements. Not only is it undirected, as noted by the authors, but it can also be difficult to discriminate between units with this measurement. But I wonder whether the authors have tested the discriminatory power of their chosen metrics? If not, a test of this could be very useful for informing us on the robustness of the text- similarity layer.

This is a great observation.

We did originally construct cosine similarity text-based networks and used QAP models that symmetrized the citations between countries (i.e., the average of the citations received, and the citations sent, the sum of the citations sent between countries, the

maximum of the cited or citing edge weight, etc.). Generally, our main findings remain steadfast, however since cosine similarity does not allow of asymmetry, these text measures are not as informative as the KLD.

Reviewer 2

1. Is this method designed to estimate country-related citation bias? Although the language is a bit ambiguous on this matter, it seems that the authors think of their approach as a method for measuring the level of citation bias in favor of core scientific nations. Specifically, they refer to the “uncomfortable spaces” in science, where citation networks and textual similarity are not aligned and to the “citational distortion” that will “continue to impose limits on the circulation of knowledge, novel ideas and future innovations”. They also talk about countries as being over-recognized and under-recognized. [...] While I commend the authors for putting the important question of global citation disparities on the scholarly agenda, I do not think their method will allow us to make any claims regarding country-level citation biases. I strongly encourage the authors to be clearer and more explicit about what they can and cannot conclude based on this approach. At its core this is a question of measurement validity. I miss a clearer statement on what it is this method actually measures and what it does not measure.

We thank Reviewer 2 for their great comment. We have provided clearer statements in the revised draft of both what the citational well measures, and what we can and cannot conclude based on our approach. In addition, we have revamped and streamlined our analyses and results to support this change.

The method is designed to estimate the degree to which the flow of citations between countries mirrors how much scientists within those same country dyads are working on similar topics in their research (this latter concept being approximated using NL-LDA and KLD). We justify this in the text as being an issue of communication barriers, where citations will be more common (i.e., higher citation counts are more likely) when communication barriers are low (i.e., when scholars are working on similar topics in their research), and where citations will be less common when communication barriers are high. One contributor to the results is national citation bias, which carries with it important implications. As Reviewer 2 points out, though, citation bias is not the only factor contributing to our results. This is a well-considered comment, and we appreciate the reviewer for bringing attention to it. For the sake of clarity, we have re-written portions of the paper to emphasize that the method is intended to capture the extent to which

citations reflect a different level of distortion relative to what is being studied as found in the texts of scientific abstracts.

The reviewer is right that our claims need to be tempered. That said, we also included a more extensive discussion of what can be done to better approximate citation bias, as this remains a valuable target even if the issue cannot be settled with certainty. (See the comments below for more details.) In addition, we have also removed the word “acknowledgement” from the title.

We have also updated how we construct our citation network based on great comments from Reviewer 3. Our citation network is now comprised as the cumulative number of citations papers used in the similarity network received five years since its publication. (We also tested a two-year window since publication and found similar results.) So now, we more directly measure how many citations over or under countries receive based on the work that is published.

2. Is this a methods paper or an empirical paper? In the discussion section, the authors frame their paper as both a methods contribution and an empirical contribution. The paper would probably be strengthened if one of these two types of contributions was given priority throughout the paper. If this is a methods contribution, I think it would be appropriate to include a more comprehensive validation of the proposed method’s performance compared to alternative approaches. If this is primarily an empirical contribution aimed at displaying global disparities, I would like to see more considerations on how to deal with database effects, variations in citation practices across sub-fields (that are distributed unequally across nations), the potential sensitivity of the labelled LDA and the relative entropy score to enormous variations in sample sizes (N papers per country).

We thank Reviewer 2 for guiding us towards thinking more critically about our paper’s contribution.

This is a method paper supported by an extensive set of validations that, as you’ll read, each parse the potential bias from various database effects and citation construction, sensitivity of our data and NL-LDA models, and imbalance in the number of papers. That said, it is difficult to offer a comparison with alternative approaches, as (a) there are no external metrics to compare against and (b) our measurement fundamentally exists at the country-dyadic level, which is different from existing approaches.

With regard to the empirical results, we have tried to address the concerns here around the sensitivity of the analysis in a few different ways that we will outline in the subsequent comments.

Notably, these sensitivity analyses include censoring on the journals in our data, censoring on English-only versus English and translated into English abstracts, citation deflation techniques, a broadened MRQAP analysis, censoring the population of journals, and hierarchical linear models (HLMs) that regress our national centrality measures on the number of papers per country.

3. Why did the authors include non-English abstracts? In the Methods section, the authors describe that they've used a python package to translate non-English abstracts into English. I'm wondering how this may affect the validity of their findings? Is it reasonable to expect that an a German or Italian research would cite a highly (textually) similar paper in Portuguese compared to a highly similar paper in English? I don't think so. To what extent may the estimated "citation distortion" be driven by language differences. It would be great to see the same analysis done without non-English abstracts to answer this question.

Reviewer 2 highlights an excellent point about English-language biases in international scientific analyses. As the lingua franca of international science, English makes up the vast majority of the papers in the database.

Nevertheless, we restricted our abstracts used to build our NL-LDA models and corresponding citation networks to English-only abstracts. In the Supplemental Materials, we re-run most of our analyses comparing the English-only results with those that include both English abstracts and those translated into English using Google Translate.

Our main findings and conclusion remain mostly the same whether censored to just English-only language abstracts or translated abstracts were included.

4. To what extent are the main results in Figure 2 driven by database effects (including the increasing representation of countries, low-impact journals, and non-English journals in MAG over time). Figure 2a, shows a notable increase in the number of countries that are present in both the international text similarity network and the international citation network over time. Figure 2b and 2c are used to demonstrate a widening gap between core scientific nations and the "all-inclusive" country sample in terms of recognition obtained, given level of textual similarity. In this sense, these figures compare the

core sample to itself over time (it's both part of core and all-inclusive). As shown in the figures, the gap between the core sample and the full sample is much smaller in 1980 than in 2015, which makes sense, since the core-sample and the all-inclusive sample are far more likely to overlap in 1980 than they are in 2015. So is inequality really increasing, or does Figure 2B and 2C merely reflect the increasing number of non-core countries with lower-impact papers in the database over time? It would be interesting to see this analysis redone with two non-overlapping samples, core vs. non-core. The population of journals in MAG may be expected to increase considerably over time, which may partly drive the increase in country representation displayed in Figure 2A. Like Web of Science, MAG probably includes a smaller set of core (high-impact) journals in the early phase (1980), with an increasing number of peripheral (low-impact) and non-English journals included over time. This begs the question as to whether the trends observed in Figures 2b and 2c are driven by language effects (non-English papers citing English papers but not the other way around) and journal effects (a higher representation of core countries publishing in high impact journals?). It would be interesting to examine how the following adjustments may affect these trends: 1) removing non-English papers from the analysis. 2) focusing on the same “core-set” of journals for all years from 1980 to 2015.

This is another great question from Reviewer 2, one that we spent a majority of our time with addressing thoroughly, as these “database effects” (we thank Reviewer 1 for highlighting this) or inclusion/exclusion criteria may artificially be driving our results.

We checked both (1) and (2) and our findings are robust to these adjustments.

On the non-English results (1), and as we previously mention we re-run most of our analyses comparing the English-only results with those that include both English abstracts and those translated into English using Google Translate in the Supplemental Materials.

On the “core-set” of journals, we re-run most of our analyses where both citations and our corpora are censored to journals that exist every year in our data, between 1980 and 2012. These data are now included in the main text instead of the uncensored version we used previously. In the Supplemental Materials, we also compare the journal censored versus the non-journal censored results and again find that our findings remain consistent.

5. To what extent may the increasing gap observed in Figure 2B and C be driven by citation inflation. According to the citation inflation principle

(<https://www.sciencedirect.com/science/article/pii/S0048733319301003>), the average number of citations per paper will increase over time (as a result of a growing population of citing papers). This begs the question as to whether the increasing gaps observed in figure 2b and 2C are driven by citation inflation as opposed to an actual increase in the acknowledgement gap (same for Fig 4). It would be interesting to see if the results are similar, if the authors adjust for citation inflation.

We thank Reviewer 2 for their great comment.

We have implemented two versions of citation deflation: (1) the field-level measure from Petersen, et al. (2019) and (2) a country-field-level measure that extends Petersen, et al.'s field-level approach. We pick the year 2000 as our baseline year that we choose to both inflate (prior to 2000) and deflate (after 2000) citations. On our country-field-level measure, for the citation centrality measures, we take the ratio of papers that a citation receiving country has at least one author associated with it between the baseline year 2000 and the alternate year between 1980 and 2012. In the Supplemental Materials, we compare these two treatments as well as the unchanged "inflated" analyses. Our findings still hold despite these deflation metrics.

6. According to the literature, it appears that the relative entropy similarity score is insensitive to sample size (number of papers per country per year). However, I'm wondering if there's any way for the authors to verify this underlying assumption in their own study where labelled LDA and the entropy similarity score are combined. Looking at Figure 3A it seems that the huge research economies are scoring highest on the text-similarity measure and that China's upwards trend corresponds with the rapid expansion of the country's research system since the 1990s. This made me think of a possible bias, where larger research nations (with more papers per year) will score higher on the similarity measure given their size alone? It would be great if the authors could reflect a bit more on the question of sample sizes in relation to their similarity score.

We thank Reviewer 2 for this excellent point.

To test if sheer size still affects our results, in particular the citation distortion centrality results in (the old) Figure 3, we run large hierarchical linear models (HLM) that we now include in the Supplemental Materials. The HLM nests countries in disciplines. We regress the citation distortion centrality for countries in fields on three independent variables (1) the percentage of GDP allocated to R&D, (2) the number of universities the country holds in year t that are ranked in the top 50 globally, and (3) the number of papers that have at least one author affiliated with it in Field t . We

report these independent variables in terms of their z-scores adjusted within disciplines to account for differences between them. In other words, countries' independent variables are mean-centered and reported in terms of their standard deviation relative to the values for countries over time within the same field. We use these other non-database country metrics to juxtapose the variation that may be the result of the number of papers produced by each country in each field.

We find that the number of papers only accounts for a small amount of variation when we estimate an HLM for the citational well. This indicates that size of papers per country is mostly not driving the results here. For completeness, we also regress text similarity centralities in using the same HLM setup outlined for the citation distortion models. We find that the variation is our dependent variable is not captured by the number of papers produced by each country in each field.

7. On page 11 (bottom), the authors state that prominent countries like the United States, UK, Germany and China are the most central in both the centrality and citation networks. However, what about countries like Switzerland and the Netherlands, which are known to be the top global performers in terms of average citation impact per paper? On p. 11 it is also stated that "the US has an average Z-score of over 7 in 2017, meaning that the U.S. receives over 7 standard deviations more incoming citations above the mean number of incoming citations of countries across fields in 2017. However, how much larger is the US in 2017 compared to the average country in terms of papers published per year? I think reflecting more explicitly on scale in these accounts would be appropriate. Likewise, it seems relevant to emphasize that biomedical research (with a high publication output) may be proportionally larger in the U.S. than in the "average" country used as a baseline, which may skew these results (due to lack of field-normalization).

We thank Reviewer 2 for this very fair point.

We should note that while our conclusions remain the same these trends are no longer as extreme in the new updated figures. To streamline the paper, we removed the extensive plots of indegree centralities in both the citation and text similarity networks. While we no longer include (our previous) citation figure in the paper, we do include what would have been the new plot here that is built using the cumulative five-year window, using English-only papers, censored to journals that have existed since 1980, and using a citation deflation measure. This figure based on our older version of Figure 3 is below for both the text similarity networks (A) and citation networks (B).

Nevertheless, similar to some of the comments above, the results can of course be disaggregated to deal with these issues, and the aggregation can be adjusted to provide a more reasonable “average” for each country. Some of our figures already hint at this.

But this is also something of a side issue. The aggregation itself is not the novel part of our methodology. Our measure exists at the field level for country dyads.

Smaller issues:

In the abstract, it is stated that “We apply a new framework called citational lensing to identify where citations should appear between countries but are nonetheless absent given that what is embedded in their published texts is highly similar.” Please consider revising “published texts” to avoid giving the impression that these analyses were made on full text material.

We agree. This can easily cause misunderstanding. We have revised the wording to “published abstract texts.”

The following sentence is missing an “around” after the “center”: “We find that scientific communities increasingly center research from highly active countries while overlooking work from peripheral countries”

Thank you for catching this and we have corrected this mistake.

Throughout the manuscript, the authors refer to a list of “core scientific nations” (including Western Europe, Australia, New Zealand, Eastern Asia and North America). How did they authors come up with this list of core-scientific nations. And in which ways can they be considered core nations? And please list all of these countries.

We thank the reviewer for the comment. In the Results section, we now include a justification for our core versus periphery status and in the Supplemental Materials itemize all countries used in our analyses.

Reviewer 3

Major point:

It is clear that citation patterns reflect a distorted scientific landscape, where certain scientific products receive a disproportionate attention with respect to others. Moreover, it is undoubtable that historical and country-specific reasons can be identified as relevant factors which have affected citation distortion to a large extent. However, starting from this evidence the paper makes three assumptions which are implausible:

1. There exists a common profile across all papers from the same country (or field and country couple) which can be inferred.
2. That profile can be obtained by solely looking at the way papers are written which can be approximated by their abstract.
3. The profile represents a credible representation of the actual value of the scientific output of a country (or field and country couple) which can be used to evaluate distortion in scientific paper success.

While the second point is more technical, the reason why the two other assumptions are unlikely is intuitive: they do not account for the way the scientific process works. To infer distortion in the citation process, a paper should be compared with other papers regarding the same topic in the same field. Furthermore, its position in the network of citations should be accounted.

Then, assuming that the paper results are as relevant as those of other comparable papers, distortions may arise from several sources. Citation inflation in a specific field or period is one of them, country bias, as claimed by Gomez et al, is another. However, country-dependence must be distinguished by the effect of the co-authorship network in which the paper authors are embedded and by the centrality of the authors' institutions in the network of academic institutions. It is after accounting for all these measurable confounding factors (which, by the way, are very common in the literature), that other unobservable country-specific features may be addressed, for instance with an LDA model.

The authors not only ignore all these alternative explanations but use a process of construction of their dependent variable (the similarity network obtained via LDA) which artificially collapse all the differences on the country dimension, hence selecting ex-ante what results they will observe.

For this reason, I do not recommend this paper for publication.

Many thanks are due to Reviewer 3 for their critical engagement with our paper. We very much appreciate the thorough read and raising many of these points. At a very high level, we have revamped and streamlined our analyses and our results, performed several robustness checks specifically targeted towards accounting for at least some of the issues Reviewer 3 raises.

Reviewer 3 seems to have a fundamental disagreement about the appropriate level of analysis. We certainly have no objection to an approach that focuses on paper-level measures. But we also think that our approach carries some very specific virtues: we are able to identify the distortion between any two countries, in any country, and in either direction. This is inherently difficult to do at the paper-level, or indeed, with the alternative approach that Reviewer 3 suggests. Our methodological approach provides unique information that can be used alongside more conventional analyses, like the one suggested by Reviewer 3. We have worked to re-emphasize this point in our revised manuscript in hopes of clarifying the unique contribution.

We do still take Reviewer 3's comments about confounding factors seriously. In the revisions here we have incorporated what we can, based on what exists at the level of analysis our method operates on (e.g., citation inflation). And we have laid out a strategy for how researchers can incorporate other factors into the analysis, as necessary. This is in keeping with the spirit of Reviewer 3's criticisms,

even if they are correct in pointing out that our method makes it difficult to incorporate paper- and researcher-level factors like citation network position and co-authorship network position.

However, mixture models are extremely common tools for characterizing populations, so we do disagree with Reviewer 3's dismissal of the NL-LDA's potential usefulness at the national-level. (For completeness, we should note that the NL-LDA model is a supervised model that is distinct from its progenitor the unsupervised LDA model.) It is perfectly fair, though, to wonder how the country-profile should be understood. The country-profile that we identify using the NL-LDA is not intended to indicate that every paper in a country shares the same profile, but that the underlying topic profile within a country affects the likelihood that papers from those countries cite one another. Whether this holds is an empirical question, of course, though it is supported by the repeatedly observed correlation between textual similarity and citation. Indeed, national boundaries are among the most important divisions in how science is organized and models and analyses are commonly performed at the national-level for this reason. Nevertheless, we have added a note highlighting that fact.

Minor points

- The paper is badly written. There are several passages which are hard to understand and need to be rewritten in a clearer manner. Oftentimes colloquial English is used. There are several typos.
- The literature review is approximate: many passages, especially the more crucial ones, are not sustained by relevant references. Moreover, sometimes the authors' claims are vague and too general.

We thank the reviewer for bringing these two issues to our attention.

Building off of the comments and observations from Reviewers 1 and 2 as well, we have revised several passages to address these concerns, including fixing typos, more formal language, and making our claims clearer and more specific.

- The plots are not appropriate. First, they are not well explained in the captions. Some (non-exhaustive) examples: annotations in Figure 1 are completely ignored in the caption; the caption in Figure 2 does not say that Figure 2A represents violin plots, does not explain what is "All" and what is "Core", does not explain what are the vertical lines in all the plot.

Furthermore, the plots sometimes are unnecessarily complicated (what is the reason for the noisy black area in Figure 4A and 4C?) and sometimes repeat useless details (differences among countries and fields are very hard to grasp in Figure 3).

This is another useful comment. We have reorganized the figures in our paper and we also clarify this in the paper. We have revamped nearly all of our figures and analyses to streamline the paper. For instance, to make the figures and annotations clearer and less redundant, we have clarified the distinction between “All” and “Core” in Figure 2 in the Results section (e.g., we now refer to “All” as “Core+Periphery”), we removed the individual lines creating the “noisy black areas” across all of our figures. We also removed several redundant figures, specifically the (previous) Figure 3 with the indegrees of both the text similarity networks and the citation networks. The vertical lines in (the previous) Figure 4A and (the previous) Figure 4C are the years 2000 and 2012, the two years we use in our nation-specific comparative cases; however, they have been removed.

- The multiplex modeling is not useful for the analysis or the discussion of results and the relation between the similarity and the citation networks could be explained without introducing the complexity of a multiplex representation.

We thank the reviewer for their comment.

Our method is specifically about characterizing the flow of information with respect to two different types of dyadic links, one represented by the citation network, and one defined by the similarity of the research topics pursued in each pair of countries, as revealed by NL-LDA and KLD. The nodes in our networks remain the same, while they are linked in different ways according to the type of connection (citation vs. textual similarity).

As such, and following any conventional definition, what we have in this paper is a multiplex network. We believe that the fundamental contribution of the method is that it provides us with distortion flows at the level of country-dyads. We believe it is more natural to use network terminology in framing the method given where it ends up. Indeed, while removing the notion of a multiplex network entirely from the paper is of course an option, we believe such an approach does not really simplify the matter and instead trades one form of complexity for another.

Decision Letter, first revision:

7th December 2021

Dear Dr. Gomez,

Thank you for submitting your revised manuscript "Rising Global Inequality in Scientific Research." (NATHUMBEHAV-210214397A). It has now been seen by the original referees and their comments are below. As you can see, the reviewers find that the paper has improved in revision. We will therefore be happy in principle to publish it in Nature Human Behaviour, pending minor revisions to satisfy the referees' final requests and to comply with our editorial and formatting guidelines.

We are now performing detailed checks on your paper and will send you a checklist detailing our editorial and formatting requirements within two weeks. Please do not upload the final materials and make any revisions until you receive this additional information from us.

Sincerely,

Charlotte Payne

Charlotte Payne, PhD
Senior Editor
Nature Human Behaviour

Reviewer #1 (Remarks to the Author):

The authors have made substantial changes and improvements to the manuscript, which fully address my own concerns. I find this to be an incredibly interesting and useful study with a truly novel methodology.

Reviewer #2 (Remarks to the Author):

Thank you for giving me the chance to revisit this manuscript. The authors have sufficiently addressed my comments, and I think the manuscript is suitable for publication in Nature Human Behaviour with a few revisions.

- 1) Sentence 2 in the abstract is difficult to comprehend and the key contribution could be conveyed more elegantly in this section.
- 2) The current structure of the paper is a bit confusing. The results section starts with several methods-heavy points and also includes points that belong in the discussion section.

3) The plotted effects in panel 2b and 3b are quite modest with an average increase/decrease in citational distortion of 0.05 over thirty years. This should probably be noted, as it demonstrates that a few countries (highlighted in the other figures) are indeed the sole winners here.

4) It's not obvious to me why Figures 2 and 3 would only include beta coefficients that are statistically significant? Why not use all of the information/variation? You're pooling these into grand averages anyway.

5) Some of the new robustness checks are not mentioned in the actual manuscript, e.g. Tables S3 and S4. The manuscript could be improved by integrating these robustness checks into the actual text. For instance, according to Table S3, adjusting for N papers reduces the variance in country-related text-similarity from 0.16 to 0.13 – that's a 19% reduction. Likewise adjusting for N-papers reduced the country-related variance in citational distortion from 0.08 to 0.07 – that's a reduction of 12.5% percent. While the mediating effect of N-papers are indeed modest, it seems relevant to highlight reductions of 19% and 12.5% attributable to size - at least if the goal is to develop a new reliable method.

Author Rebuttal, first revision:

The reviewers' comments are in normal text and **our responses are in bold and bordered.**

Reviewer 1

The authors have made substantial changes and improvements to the manuscript, which fully address my own concerns. I find this to be an incredibly interesting and useful study with a truly novel methodology.

We thank Reviewer 1 for all their diligence throughout this review process. Their notes have substantially improved the paper.

Reviewer 2

Thank you for giving me the chance to revisit this manuscript. The authors have sufficiently addressed my comments, and I think the manuscript is suitable for publication in Nature Human Behaviour with a few revisions.

We thank you for all the recommendations and suggestions through this process. Indeed,

the paper is all that much better because of them.

1. Sentence 2 in the abstract is difficult to comprehend and the key contribution could be conveyed more elegantly in this section.

We agree. The abstract has been updated.

2. The current structure of the paper is a bit confusing. The results section starts with several methods-heavy points and also includes points that belong in the discussion section.

This has been fixed. Those sections in the Results section that were very methods-heavy are now in the methods-section, and points that ought to belong in the Discussion section can now be found there.

3. The plotted effects in panel 2b and 3b are quite modest with an average increase/decrease in citational distortion of 0.05 over thirty years. This should probably be noted, as it demonstrates that a few countries (highlighted in the other figures) are indeed the sole winners here.

We thank Reviewer 2 for their comment. We agree. We have added this point to the Discussion section.

“Indeed, as demonstrated in Figures 3A, 3C, and 4, only a few highlighted countries are the real sole winners here, as the overall average beta-coefficient trend plotted in Figure 2B and the overall average citational distortion trend plotted in 3B are quite modest over thirty years.

4. It's not obvious to me why Figures 2 and 3 would only include beta coefficients that are statistically significant? Why not use all of the information/variation? You're pooling these into grand averages anyway.

This is a great point. Here, we argue that pooling statistically significant betas are a more

rigorous approach, but we also agree that the beta coefficients are not the unit of comparison for the Figures. To this end, we have included a new section in the Supplemental Materials that plot Figure 2 with all beta coefficients irrespective of whether they are statistically significant or not. (Figure 3 did not plot beta coefficients.) Figures 2B and 2C with all betas can be found in SM Figure S30 and Figure S31, respectively.

5. Some of the new robustness checks are not mentioned in the actual manuscript, e.g. Tables S3 and S4. The manuscript could be improved by integrating these robustness checks into the actual text. For instance, according to Table S3, adjusting for N papers reduces the variance in country-related text-similarity from 0.16 to 0.13 – that’s a 19% reduction. Likewise adjusting for N-papers reduced the country-related variance in citational distortion from 0.08 to 0.07 – that’s a reduction of 12.5% percent. While the mediating effect of N-papers are indeed modest, it seems relevant to highlight reductions of 19% and 12.5% attributable to size - at least if the goal is to develop a new reliable method.

Thank you for this comment. We now reference the results of Table S3 and Table S4 in the Results subsection “Citations and Recognition.”

“(In the Supplemental Materials, Tables S3 and S4 in the subsection Hierarchical Linear Models (HLMs), we test how much variance in our citational distortion and thus our text similarity measures are due to the result of the sheer volume of papers produced by authors from countries using hierarchical linear models. The mediating impact of N papers is modest. Adjusting for N papers reduces the variance in country-related text-similarity by 19%, from 0.16 to 0.13. Likewise, adjusting for N papers reduced the country-related variance in citational distortion by 12.5% from 0.08 to 0.07.)”

Final Decision Letter: